# A Batch-to-Online Transformation under Random-Order Model

**Jing Dong**
The Chinese University of Hong Kong, Shenzhen
jingdong@link.cuhk.edu.cn

**Yuichi Yoshida**
National Institute of Informatics
yyoshida@nii.ac.jp

## Abstract

We introduce a transformation framework that can be utilized to develop online algorithms with low $\epsilon$-approximate regret in the random-order model from offline approximation algorithms. We first give a general reduction theorem that transforms an offline approximation algorithm with low average sensitivity to an online algorithm with low $\epsilon$-approximate regret. We then demonstrate that offline approximation algorithms can be transformed into a low-sensitivity version using a coreset construction method. To showcase the versatility of our approach, we apply it to various problems, including online $(k, z)$-clustering, online matrix approximation, and online regression, and successfully achieve polylogarithmic $\epsilon$-approximate regret for each problem. Moreover, we show that in all three cases, our algorithm also enjoys low inconsistency, which may be desired in some online applications.

## 1 Introduction

In online learning literature, stochastic and adversarial settings are two of the most well-studied cases. Although the stochastic setting is not often satisfied in real applications, the performance and guarantees of online algorithms in the adversarial case are considerably compromised. This is particularly true for important online tasks such as $k$-means clustering, which gives a significantly worse guarantee than their offline or stochastic counterparts [Cohen-Addad et al., 2021]. As a result, their practical applicability is greatly limited.

Recently, the random-order model has been introduced as a means of modeling learning scenarios that fall between the stochastic and adversarial settings [Garber et al., 2020, Sherman et al., 2021]. In the random-order model, the adversary is permitted to choose the set of losses, with full knowledge of the learning algorithm, but has no influence over the order in which the losses are presented to the learner. Instead, the loss sequence is uniformly and randomly permuted. This effectively bridges the gap between the stochastic setting, where only the distribution of losses can be chosen by the setting, and the adversarial setting, where the adversary has complete control over the order of the losses presented to the learner.

In this work, we introduce a batch-to-online transformation framework designed specifically for the random-order model. Our framework facilitates the conversion of an offline approximation algorithm into an online learning algorithm with $\epsilon$-approximate regret guarantees. Our primary technical tool is average sensitivity, which was initially proposed by Varma and Yoshida [2021] to describe the algorithm's average-case sensitivity against input perturbations. We demonstrate that any offline approximation algorithm with low average sensitivity will result in a transformed online counterpart that has low $\epsilon$-approximate regret. To achieve small average sensitivity for offline algorithms, we leverage the idea of a coreset [Agarwal et al., 2005, Har-Peled and Mazumdar, 2004], which is a small but representative subset of a larger dataset that preserves important properties of the original data. We present a coreset construction method that attains low average sensitivity, and when combined with the approximation algorithm, yields an overall algorithm with low average sensitivity.

37th Conference on Neural Information Processing Systems (NeurIPS 2023).

To showcase the practicality and versatility of our framework, we apply it to three popular online learning problems: online $(k, z)$-clustering, online matrix approximation, and online regression. In all three cases, our approach yields a polylogarithmic $\epsilon$-approximate regret. Furthermore, due to the low average sensitivity of our algorithms, they also enjoy low inconsistency, which is the cumulative number of times the solution changes. This additional property may prove useful in certain online settings. We note that this inconsistency has also been investigated in the classic online learning and multi-armed bandits literature [Agrawal et al., 1988, Cesa-Bianchi et al., 2013].

## 2 Related Works

**Average sensitivity** Varma and Yoshida [2021] first introduced the notion of average sensitivity and proposed algorithms with low average sensitivity on graph problems such as minimum spanning tree, minimum cut, and minimum vertex cover problems. Various other problems have then been analyzed for the average sensitivity, including dynamic programming problems [Kumabe and Yoshida, 2022], spectral clustering [Peng and Yoshida, 2020], Euclidean $k$-clustering [Yoshida and Ito, 2022], maximum matching problems [Yoshida and Zhou, 2021], and decision tree learning problem [Hara and Yoshida, 2023].

**Online (consistent) $(k, z)$-clustering** While $(k, z)$-clustering, which includes $k$-means ($z = 2$) and $k$-median ($z = 1$) as its special cases, has been studied extensively from various perspectives such as combinatorial optimization and probabilistic modeling, it can be NP-hard to obtain the exact solution [Impagliazzo et al., 2001]. Thus most theoretical works have been focused on designing approximation algorithms. In the online setting, Li et al. [2018] proposed a Bayesian adaptive online clustering algorithm that enjoys a minimal sublinear regret. However, the algorithm is allowed to output more than $k$ clusters. Without such assumption, Cohen-Addad et al. [2021] proposed the first algorithm that attains $\epsilon$-approximate regret of $O(k\sqrt{d^3 n} \log(\epsilon^{-1} dkn))$ for $k$-means clustering under adversarial setting.

On a separate vein, Lattanzi and Vassilvitskii [2017] proposed an online consistent $(k, z)$-clustering algorithm that produces a $2^{O(z)}$-approximate solution for the data points obtained so far at each step while maintaining an inconsistency bound of $O(k^2 \log^4 n)$. This implies that their algorithm only updates the output $O(k^2 \log^4 n)$ many times. Then, Yoshida and Ito [2022] gave an online algorithm with approximation ratio $(1 + \epsilon)$ and inconsistency $\text{poly}(d, k, 2^z, \epsilon^{-1}) \cdot \log n$ in the random-order model. We remark that the way how the losses are computed in Lattanzi and Vassilvitskii [2017], Yoshida and Ito [2022] is different from that of the online setting, which Cohen-Addad et al. [2021] and our paper considered.

**Online convex optimization and online principle component analysis (PCA) under the random-order model** The online random-order optimization was proposed in Garber et al. [2020], which established a bound of $O(\log n)$ for smooth and strongly convex losses. This result is then improved by Sherman et al. [2021] while still requiring smooth and convex losses.

The techniques and results are then extended to online PCA with the random-order setting, for which a regret of $O\left(\zeta^{-1}\sqrt{kn}\right)$ was established, where $\zeta$ is an instance-dependent constant. This recovers the regret for online PCA in the stochastic setting [Warmuth and Kuzmin, 2008, Nie et al., 2016]. We remark that PCA can be viewed as a special case of matrix approximation, in which the matrix being approximated is the covariance matrix of the data, and we discuss the more general problem of matrix approximation in this paper.

## 3 Preliminaries

For a positive integer $n$, let $[n]$ denote the set $\{1, 2, \ldots, n\}$. For real values $a, b \in \mathbb{R}$, $a \in (1 \pm \epsilon)b$ is a shorthand for $(1 - \epsilon)b \leq a \leq (1 + \epsilon)b$.

### 3.1 Offline Learning

We consider a general class of learning problems. Let $\mathcal{X}$ be the input space, $\Theta$ be the parameter space, and $\ell : \Theta \times \mathcal{X} \to \mathbb{R}_+$ be a loss function. For simplicity, we assume the loss is bounded, i.e.,

$\ell(\theta, x) \leq 1$. Given a set of $n$ data points $X \in \mathcal{X}^n$, we are asked to learn a parameter $\theta \in \Theta$ that minimizes the objective value $\ell(\theta, X) := \sum_{x \in X} \ell(\theta, x)$. We call this problem the *offline learning problem*.

When the exact minimization of the loss function $\ell$ is NP-hard or computationally demanding, one may only hope to obtain an approximate solution efficiently. Specifically, for $\alpha > 0$, we say a solution $\theta \in \Theta$ is $\alpha$-*approximate* for $X \in \mathcal{X}^n$ if $\ell(\theta, X) \leq \alpha \cdot \min_{\tilde{\theta} \in \Theta} \ell(\tilde{\theta}, X)$. The value $\alpha$ is called the *approximation ratio* of the solution. We say a (possibly randomized) algorithm $\mathcal{A}$ is $\alpha$-*approximate* if the expected approximation ratio of the output solution is at most $\alpha$.

### 3.2 Online Learning with Random-Order Model

In the *online learning problem*, instead of receiving all points at once, the data arrives sequentially throughout a time horizon $n$. Specifically, the data point comes one by one, where $x_t$ comes at time $t \in [n]$. At the time $t$, using the collected data points $X_{t-1} := (x_1, \ldots, x_{t-1})$, we are asked to output a parameter $\theta_t \in \Theta$. Then we receive the data point $x_t$ and incur a loss of $\ell(\theta_t, x_t)$. In this work, we consider the *random-order model*, in which the data points $x_1, \ldots, x_n$ may be chosen adversarially, but their ordering is randomly permuted before the algorithm runs.

To evaluate our performance, we use the notion of regret, which is the cumulative difference between our solution and the best solution in hindsight. In cases where obtaining the exact solution is hard, and one may only hope to obtain an approximate solution efficiently, we use the $\epsilon$-*approximate regret*.

**Definition 3.1** ($\epsilon$-approximate regret for the random-order model)**.** *Given a (randomized) algorithm $\mathcal{A}$ that outputs a sequence of parameters $\theta_1, \ldots, \theta_n$ when given input $x_1, \ldots, x_n$. The $\epsilon$-approximate regret of $\mathcal{A}$ for the random-order model is defined as*

$$\mathrm{Regret}_\epsilon(n) := \mathop{\mathbb{E}}_{\mathcal{A}, \{x_t\}} \left[ \sum_{t=1}^n \ell(\theta_t, x_t) - (1 + \epsilon) \cdot \min_{\tilde{\theta} \in \Theta} \sum_{t=1}^n \ell(\tilde{\theta}, x_t) \right].$$

*where the randomness is over the internal randomness of $\mathcal{A}$ and the ordering of data points. When $\epsilon = 0$, we simply call it the* regret.

In certain cases, online algorithms are required to maintain a good solution while minimizing *inconsistency*, which is quantified as the number of times the solution changes. This can be expressed formally as $\mathrm{Inconsistency}(n) = \mathbb{E}_{\mathcal{A}, \{x_t\}}[\sum_{t=1}^{n-1} \mathbb{I}\{\theta_t \neq \theta_{t+1}\}]$, where $\mathbb{I}$ is the indicator function.

### 3.3 Average sensitivity

On a high level, the notion of average sensitivity describes the differences in the performance of a randomized algorithm with respect to input changes. This difference is captured by the total variation distance, which is defined below.

**Definition 3.2.** *For a measurable space $(\Omega, \mathcal{F})$ and probability measures $P, Q$ defined on $(\Omega, \mathcal{F})$. The total variation distance between $P$ and $Q$ is defined as $\mathrm{TV}(P, Q) := \sup_{A \in \mathcal{F}} |P(A) - Q(A)|$.*

Equipped with this, the average sensitivity of a randomized algorithm is formally defined as the average total variation distance between the algorithm's output on two training data sets that differ by deleting one point randomly. For a dataset $X = (x_1, \ldots, x_n) \in \mathcal{X}^n$ and $i \in [n]$, let $X^{(i)}$ denote the set $(x_1, \ldots, x_{i-1}, x_{i+1}, \ldots, x_n)$ obtained by deleting the $i$-th data point. Then, the following definition gives a detailed description of the notion:

**Definition 3.3** (Average Sensitivity [Varma and Yoshida, 2021, Yoshida and Ito, 2022])**.** *Let $\mathcal{A}$ be a (randomized) algorithm that takes an input $X \in \mathcal{X}^n$ and outputs $\mathcal{A}(X)$. For $\beta : \mathbb{Z}_+ \to \mathbb{R}_+$, we say that the average sensitivity of $\mathcal{A}$ is at most $\beta$ if*

$$\frac{1}{n} \sum_{i=1}^n \mathrm{TV}(\mathcal{A}(X), \mathcal{A}(X^{(i)})) \leq \beta(n),$$

*for any $X \in \mathcal{X}^n$, where we identify $\mathcal{A}(X)$ and $\mathcal{A}(X^{(i)})$ with their distributions.*

## 4 Batch-to-Online Transformation in the Random-Order Model

In this section, we describe a general framework that can transform any offline $(1 + \epsilon)$-approximate algorithm into an online algorithm with low $\epsilon$-approximate regret. Our goal is to show the following.

**Theorem 4.1.** *Let $\mathcal{A}$ be a (randomized) $(1 + \epsilon)$-approximate algorithm for the offline learning algorithm with average sensitivity $\beta : \mathbb{Z}_+ \to \mathbb{R}_+$. Then, there exists an online learning algorithm in the random-order model such that* $\mathrm{Regret}_\epsilon(n) = O\left(\sum_{t=1}^n \beta(t) + 1\right)$.

Our method is described in Algorithm 1. Let $\mathcal{A}$ be an approximation algorithm for the offline learning problem. Then, at each time step, based on the collected data $X_{t-1}$, we simply output $\theta_t = \mathcal{A}(X_{t-1})$.

---

**Algorithm 1:** General batch-to-online conversion

**Input:** Offline approximation algorithm $\mathcal{A}$.

1 **for** $t = 1, \dots, n$ **do**
2 $\quad$ Obtain $\theta_t$ by running $\mathcal{A}$ on $X_{t-1}$.
3 $\quad$ Receive $x_t$ and $\ell(\theta_t, x_t)$.

---

To show that Algorithm 1 achieves a low approximate regret when $\mathcal{A}$ has a low average sensitivity, the following lemma is useful.

**Lemma 4.2.** *Let $\mathcal{A}$ be a (randomized) algorithm for the offline learning problem with average sensitivity $\beta : \mathbb{Z}_+ \to \mathbb{R}_+$. Then for any input $X \in \mathcal{X}^n$, we have*

$$\frac{1}{n} \sum_{i=1}^n \mathbb{E}_{\mathcal{A}}[\ell(\mathcal{A}(X^{(i)}), x_i)] = \frac{1}{n} \sum_{i=1}^n \mathbb{E}_{\mathcal{A}}[\ell(\mathcal{A}(X), x_i)] \pm \beta(n),$$

*where $x = a \pm b$ means $a - b \le x \le a + b$.*

*Proof of Theorem 4.1.* Consider Algorithm 1. For any $t \in [n]$, we have

$$\mathbb{E}_{\mathcal{A},\{x_i\}} \left[ \ell(\theta_{t+1}, x_{t+1}) - \frac{1}{t}\ell(\theta_{t+1}, X_t) \right] = \mathbb{E}_{\mathcal{A},\{x_i\}} \left[ \frac{1}{t} \sum_{i=1}^t (\ell(\theta_{t+1}, x_{t+1}) - \ell(\theta_{t+1}, x_i)) \right]$$

$$= \mathbb{E}_{\mathcal{A},\{x_i\}} \left[ \frac{1}{t} \sum_{i=1}^t (\ell(\mathcal{A}(X_t), x_{t+1}) - \ell(\mathcal{A}(X_t), x_i)) \right]$$

$$\le \mathbb{E}_{\mathcal{A},\{x_i\}} \left[ \frac{1}{t} \sum_{i=1}^t \left( \ell(\mathcal{A}(X_t), x_{t+1}) - \ell(\mathcal{A}(X_t^{(i)}), x_i) \right) \right] + \beta(t) \qquad \text{(By Lemma 4.2)}$$

$$= \mathbb{E}_{\mathcal{A},\{x_i\}} \left[ \frac{1}{t} \sum_{i=1}^t \left( \ell(\mathcal{A}(X_t), x_{t+1}) - \ell(\mathcal{A}(X_t^{(i)}), x_{t+1}) \right) \right] + \beta(t)$$

$$\le \mathbb{E}_{\mathcal{A},\{x_i\}} \left[ \frac{1}{t} \sum_{i=1}^t \mathrm{TV}(\mathcal{A}(X_t), \mathcal{A}(X_t^{(i)})) \right] + \beta(t) \le 2\beta(t),$$

where the last equality follows by replacing $x_i$ with $x_{t+1}$ in $\ell(\mathcal{A}(X_t^{(i)}), x_i)$ because they have the same distribution, and the last inequality is by the average sensitivity of the algorithm.

Rearranging the terms, we have

$$\mathbb{E}_{\mathcal{A},\{x_i\}} [\ell(\theta_{t+1}, x_{t+1})] \le \mathbb{E}_{\mathcal{A},\{x_i\}} \left[ \frac{\ell(\theta_{t+1}, X_t)}{t} \right] + 2\beta(t) \le \mathbb{E}_{\{x_i\}} \left[ \frac{(1+\epsilon)\mathrm{OPT}_t}{t} \right] + 2\beta(t),$$

where $\mathrm{OPT}_t := \min_\theta \ell(\theta, X_t)$ is the optimal value with respect to $X_t$, and the second inequality holds because the approximation ratio of $\theta_{t+1}$ is $1 + \epsilon$ in expectation.

Taking summation over both sides, we have

$$\mathbb{E}_{\mathcal{A},\{x_i\}} \left[ \sum_{t=1}^n \ell(\theta_t, x_t) \right] = \mathbb{E}_{\mathcal{A},\{x_i\}} [\ell(\theta_1, x_1)] + \mathbb{E}_{\mathcal{A},\{x_i\}} \left[ \sum_{t=1}^{n-1} \ell(\theta_{t+1}, x_{t+1}) \right]$$

$$\leq 1 + \mathop{\mathbb{E}}_{\{x_i\}} \left[ \sum_{t=1}^{n-1} \frac{(1+\epsilon)\mathrm{OPT}_t}{t} \right] + 2 \sum_{t=1}^{n-1} \beta(t) .$$

Fix the ordering $x_1, \ldots, x_n$, and let $c_i$ $(i \in [t])$ be the loss incurred by $x_i$ in $\mathrm{OPT}_n$. In particular, we have $\mathrm{OPT}_n = \sum_{i=1}^{n} c_i$. Note that $c_i$'s are random variables depending on the ordering of data points, but their sum, $\mathrm{OPT}_n$, is deterministic. Then, we have $\mathrm{OPT}_t \leq \sum_{i=1}^{t} c_i$ because $\mathrm{OPT}_t$ minimizes the loss up to time $t$, Hence, we have

$$\mathop{\mathbb{E}}_{\{x_i\}} \left[ \sum_{t=1}^{n} \frac{\mathrm{OPT}_t}{t} \right] \leq \mathop{\mathbb{E}}_{\{x_i\}} \left[ \sum_{t=1}^{n} \frac{\sum_{i=1}^{t} c_i}{t} \right] = \mathop{\mathbb{E}}_{\{x_i\}} \left[ \sum_{i=1}^{n} c_i \sum_{t=i}^{n} \frac{1}{t} \right] = \sum_{i=1}^{n} \mathop{\mathbb{E}}_{\{x_i\}} [c_i] \sum_{t=i}^{n} \frac{1}{t}$$

$$= \frac{\mathrm{OPT}_n}{n} \cdot \sum_{i=1}^{n} \sum_{t=i}^{n} \frac{1}{t} = \frac{\mathrm{OPT}_n}{n} \cdot n = \mathrm{OPT}_n .$$

Therefore, we have

$$\mathop{\mathbb{E}}_{\mathcal{A}, \{x_i\}} \left[ \sum_{t=1}^{n} \ell(\theta_t, x_t) \right] - (1+\epsilon)\mathrm{OPT}_n = O \left( \sum_{t=1}^{n} \beta(t) + 1 \right) . \qquad \square$$

## 5 Approximation Algorithm with Low Average Sensitivity via Coreset

To design approximation algorithms for the offline learning problem with low average sensitivity, we consider the following approach: We first construct a small subset of the input that well preserves objective functions, called a coreset, with small average sensitivity, and then apply any known approximation algorithm on the coreset. Coreset is formally defined as follows:

**Definition 5.1** (Har-Peled and Mazumdar [2004], Agarwal et al. [2005]). *Let $\ell : \Theta \times \mathcal{X} \to \mathbb{R}_+$ be a loss function and let $X \in \mathcal{X}^n$. For $\epsilon > 0$, we say that a weighted set $(Y, w)$ with $Y \subseteq X$ and $w : Y \to \mathbb{R}_+$ is an $\epsilon$-coreset of $X$ with respect to $\ell$ if for any $\theta \in \Theta$, we have $\sum_{y \in Y} w(y)\ell(\theta, y) \in (1 \pm \epsilon) \sum_{x \in X} \ell(\theta, x)$.*

Now, we consider a popular method for constructing coresets based on importance sampling and show that it enjoys a low average sensitivity. For a data $x \in X$, its *sensitivity* $\sigma_X(x)$[1] is its maximum contribution to the loss of the whole dataset, or more formally

$$\sigma_X(x) = \sup_{\theta \in \Theta} \frac{\ell(\theta, x)}{\ell(\theta, X)} . \tag{1}$$

---

**Algorithm 2:** Coreset Construction Based on Sensitivity Sampling

**Input:** Loss function $\ell : \Theta \times \mathcal{X} \to \mathbb{R}_+$, dataset $X \in \mathcal{X}^n$, $m \in \mathbb{N}$, and $\epsilon > 0$

1 For each $x \in X$, compute $\sigma_X(x)$ and set $p(x) = \sigma_X(x) / \sum_{x' \in X} \sigma_X(x')$.
2 Let $S$ be an empty set.
3 **for** $i = 1, \ldots, m$ **do**
4      Sample $x$ with probability $p(x)$.
5      Sample $\tilde{p}$ from $[p(x), (1 + \epsilon/2)p(x)]$ uniformly at random.
6      **if** $w(x)$ *is undefined* **then**
7          $S \leftarrow S \cup \{x\}$.
8          $w(x) \leftarrow 1/\tilde{p}$.
9      **else**
10          $w(x) \leftarrow w(x) + 1/\tilde{p}$.
11      **return** $(S, w)$.

---

It is known that we can construct a coreset as follows: A data point $x \in X$ is sampled with probability $p(x) := \sigma_X(x) / \sum_{x' \in X} \sigma_X(x')$, and then its weight in the output coreset is increased by $1/\tilde{p}$, where

---

[1]The reader should not confuse sensitivity, which is a measure for data points, with average sensitivity, which is a measure for algorithms.

$\tilde{p}$ is a slight perturbation of $p(x)$. This process is to be repeated for a fixed number of times, where the exact number depends on the approximation ratio of the coreset. See Algorithm 2 for details. We can bound its average sensitivity as follows:

**Lemma 5.2.** *The average sensitivity of Algorithm 2 is* $O\left(\epsilon^{-1}m/n\right)$.

A general bound on the number of times we need to repeat the process, i.e., $m$ in Algorithm 2, to obtain an $\epsilon$-coreset is known (see, e.g., Theorem 5.5 of Braverman et al. [2016]). However, we do not discuss it here because better bounds are known for specific problems and we do not use the general bound in the subsequent sections.

# 6 Online $(k, z)$-Clustering

In online applications, unlabelled data are abundant and their structure can be essential, and clustering serves as an important tool for analyzing them. In this section, as an application of our general batch-to-online transformation, we describe an online $(k, z)$-clustering method that enjoys low regret.

## 6.1 Problem setup

The online $(k, z)$-clustering problem [Cohen-Addad et al., 2021] is an instance of the general online learning problem described in Section 3. We describe the problem as follows: Let $k \geq 1$ be an integer and $z \geq 1$ be a real value. Over a time horizon $n$, at each time step $t$, a data point $x_t \in \mathbb{R}^d$ is given. Using the set of data points $X_{t-1} = \{x_1, \ldots, x_{t-1}\}$, we are asked to compute a set $Z_t = \{z_1, \ldots, z_k\}$ of $k$ points in $\mathbb{R}^d$ that minimize $\ell(Z_t, x_t) := \min_{j=1,\ldots,k} \|x_t - z_j\|_2^z$, which is the $z$-th power of the Euclidean distance between $x_t$ and the closest point in $Z_t$. Note that $Z_t$ plays the role of $\theta_t$ in the general online learning problem. The regret and $\epsilon$-approximate regret are defined accordingly.

## 6.2 Method and results

One important ingredient to our method is the coreset construction method proposed by Huang and Vishnoi [2020]. The method provides a unified two-stage importance sampling framework, which allows for a coreset with a size that is dimension independent. Specifically, the method constructs an $\epsilon$-coreset of size $\tilde{O}\left(\min\left\{\varepsilon^{-2z-2}k, 2^{2z}\epsilon^{-4}k^2\right\}\right)$ in $\tilde{O}(ndk)$ time, where the $\tilde{O}$ hides polylogarithmic factors in $n$ and $k$. We remark that the importance of sampling steps in the framework is similar to the ones described in Section 5, which thus allows us to analyze its average sensitivity.

Algorithm 3 gives a brief description of our algorithm, while a detailed description is presented in the appendix. The algorithm adheres to the standard transformation approach, whereby an offline approximation algorithm is run on the coreset derived from the aggregated data.

---

**Algorithm 3:** Online consistent $(k, z)$-clustering

---

**Input:** Offline algorithm $\mathcal{A}$ for $(k, z)$-clustering, approximation ratio $1 + \epsilon$, $\epsilon \in (0, 1)$.

1   $\epsilon' \leftarrow \epsilon/3$.
2   **for** $t = 1, \ldots, n$ **do**
3      Construct an $\epsilon'$-coreset $C_{t-1} = (S_{t-1}, \omega_{t-1})$ on $X_{t-1}$.
4      Obtain a cluster set $Z_t$ by running a PTAS $\mathcal{A}$ with approximation ratio of $(1 + \epsilon')$ on $C_{t-1}$.
5      Receive $x_t \in \mathbb{R}^d$ and $\ell(Z_t, x_t) \in \mathbb{R}_+$.

---

**Theorem 6.1.** *For any $\epsilon \in (0, 1)$, Algorithm 3 gives a regret bound of*

$$\text{Regret}_\epsilon(n) \leq O\left(\left((168z)^{10z}\epsilon^{-5z-15}k^5 \log \frac{kn}{\epsilon} + \epsilon^{-2z-2}k \log k \log \frac{kn}{\epsilon}\right) \log n\right).$$

*Moreover, there exists an algorithm that enjoys the same regret bound and an inconsistency bound of* $\text{Inconsistency}(n) = O\left(\left((168z)^{10z}\epsilon^{-5z-15}k^5 \log(\epsilon^{-1}kn) + \epsilon^{-2z-2}k \log k \log(\epsilon^{-1}kn)\right) \log n\right)$ *for $(k, z)$-clustering.*

**Remark 6.1.** *When $z = 2$, previous results for the adversarial setting show an $\epsilon$-approximate regret bound of $O(k\sqrt{d^3n}\log(\epsilon^{-1}dkn))$ [Cohen-Addad et al., 2021]. In comparison, although our regret is for the random-order model, our method and results accommodate a range of values for $z$, and the regret bound is only polylogarithmically dependent on $n$ and is independent of the dimension $d$.*

# 7   Online Low-Rank Matrix Approximation

Low-rank matrix approximation serves as a fundamental tool in statistics and machine learning. The problem is to find a rank-$k$ matrix that approximates an input matrix $\mathbf{A} \in \mathbb{R}^{d \times n}$ as much as possible. In this section, we apply the transformation framework to the offline approximation algorithm to obtain a low regret online algorithm.

## 7.1   Problem setup

**Low-rank matrix approximation**   By the singular value decomposition (SVD), a rank-$r$ matrix $\mathbf{A} \in \mathbb{R}^{d \times n}$ can be decomposed as $\mathbf{A} = \mathbf{U}\boldsymbol{\Sigma}\mathbf{V}^\top$, where $\mathbf{U} \in \mathbb{R}^{d \times r}$ and $\mathbf{V} \in \mathbb{R}^{n \times r}$ are orthonormal matrices, $\boldsymbol{\Sigma} \in \mathbb{R}^{r \times r}$ is a diagonal matrix with $\mathbf{A}$'s singular values on the diagonal. The best rank-$k$ approximation of $\mathbf{A}$ is given by

$$\mathbf{A}_k = \mathbf{U}_k\boldsymbol{\Sigma}_k\mathbf{V}_k^\top = \underset{\mathbf{B} \in \mathbb{R}^{d \times n}:\text{rank}(\mathbf{B}) \leq k}{\text{argmin}} \|\mathbf{A} - \mathbf{B}\|_F \ ,$$

where $\|\cdot\|_F$ denotes the Frobenius norm, $\boldsymbol{\Sigma}_k \in \mathbb{R}^{k \times k}$ is a diagonal matrix with $\mathbf{A}_k$'s top $k$ singular values on the diagonal, and $\mathbf{U}_k \in \mathbb{R}^{d \times k}$ and $\mathbf{V}_k \in \mathbb{R}^{n \times k}$ are orthonormal matrices obtained from $\mathbf{U}$ and $\mathbf{V}$, respectively, by gathering corresponding columns. The best rank-$k$ approximation can also be found by projecting $\mathbf{A}$ onto the span of its top $k$ singular vectors, that is, $\mathbf{A}_k = \mathbf{U}_k\mathbf{U}_k^\top\mathbf{A}$. Then, we can say an orthonormal matrix $\mathbf{Z}$ is an $\epsilon$-approximate solution if

$$\left\|\mathbf{A} - \mathbf{Z}\mathbf{Z}^\top\mathbf{A}\right\|_F \leq (1 + \epsilon)\left\|\mathbf{A} - \mathbf{U}_k\mathbf{U}_k^\top\mathbf{A}\right\|_F \ .$$

The matrix approximation problem serves as an important tool in data analytics and is closely related to numerous machine learning methods such as principal component analysis and least squares analysis. When dealing with streaming data, the online version of the matrix approximation problem becomes a vital tool for designing online versions of the machine learning algorithms mentioned above.

**Online matrix approximation**   Through a time horizon of $n$, we receive a column of $\mathbf{A}$, $a_t \in \mathbb{R}^d$ at each time step $t$. We are then asked to compute $\mathbf{Z}_t \in \mathbb{R}^{d \times k}$ that minimizes

$$\ell(\mathbf{Z}_t, a_t) = \left\|a_t - \mathbf{Z}_t\mathbf{Z}_t^\top a_t\right\|_F \ .$$

Without loss of generality, we will assume that the losses are bounded between $[0, 1]$. We remark that similar assumptions are also made in Nie et al. [2016].

The online matrix approximation problem serves as a core component of online machine learning algorithms such as principle component analysis. These algorithms are important to a range of applications, such as online recommendation systems and online experimental design [Warmuth and Kuzmin, 2008, Nie et al., 2016].

## 7.2   Method and results

In the context of low-rank matrix approximation, the coreset of a matrix is called the projection-cost preserving samples, defined as follows:

**Definition 7.1** (Rank-$k$ Projection-Cost Preserving Sample Cohen et al. [2017])**.** *For $n' < n$, a subset of rescaled columns $\mathbf{C} \in \mathbb{R}^{d \times n'}$ of $\mathbf{A} \in \mathbb{R}^{d \times n}$ is a $(1 + \epsilon)$ projection-cost preserving sample if, for all rank-$k$ orthogonal projection matrices $\mathbf{X} \in \mathbb{R}^{d \times d}$, $(1 - \epsilon)\|\mathbf{A} - \mathbf{X}\mathbf{A}\|_F^2 \leq \|\mathbf{C} - \mathbf{X}\mathbf{C}\|_F^2 \leq (1 + \epsilon)\|\mathbf{A} - \mathbf{X}\mathbf{A}\|_F^2$.*

Such sketches that satisfy Definition 7.1 can be constructed via importance sampling-based routines, which are modifications of the "leverage scores". Specifically, for the $i$-th column $a_i$ of matrix $A$,

the *ridge leverage score* is defined as $\tau_i(\mathbf{A}) = a_i^\top \left( \mathbf{A}\mathbf{A}^\top + \frac{\|\mathbf{A}-\mathbf{A}_k\|_F^2}{k}\mathbf{I} \right)^\dagger a_i$, where $\dagger$ denotes the Moore-Penrose pseudoinverse of a matrix [Cohen et al., 2017].

Now, we introduce our online matrix approximation algorithm in Algorithm 4, which builds upon our transformation framework. It computes the approximation of the matrix from the sketch derived from the aggregated matrix using ridge leverage scores.

---

**Algorithm 4:** Online low rank matrix approximation

**Input:** Approximation parameters $\epsilon \in (0, 1)$.

1 Set $\delta = O(\epsilon/n)$ and $m = O\left(\epsilon^{-2}k\log(\delta^{-1}k)\right)$.

2 **for** $t = 1, \ldots, n$ **do**

3      Construct $\mathbf{A}_{t-1} \in \mathbb{R}^{d\times(t-1)}$ by concatenating $a_1, \ldots a_{t-1}$.

4      Let $\mathbf{C}_{t-1} \in \mathbb{R}^{d\times m}$ be the zero matrix.

5      **for** $j = 1, \ldots, m$ **do**

6          Sample the $i$-th column $a_i \in \mathbb{R}^d$ of $\mathbf{A}_{t-1}$ with probability $p_i := \frac{\tau_i(\mathbf{A}_{t-1})}{\sum_{j=1}^{t-1}\tau_j(\mathbf{A}_{t-1})}$.

7          Sample $w \in \mathbb{R}$ uniformly from $[1/\sqrt{tp_i}, (1+\epsilon)/\sqrt{tp_i}]$.

8          Replace the $j$-th column of $\mathbf{C}_{t-1}$ with $w \cdot a_i$.

9      Set $\mathbf{Z}_t \in \mathbb{R}^{d\times k}$ to the top $k$ left singular vectors of $\mathbf{C}_t$

10      Receive $a_t \in \mathbb{R}^d$ and $\ell(\mathbf{Z}_t, a_t) \in \mathbb{R}_+$.

---

**Theorem 7.2.** *For any $\epsilon \in (0,1)$, Algorithm 4 has regret $\mathrm{Regret}_\epsilon(n) = O\left(\epsilon^{-2}k\log n \log(\epsilon^{-1}kn)\right)$. Moreover, there exists an algorithm for online low-rank matrix approximation that enjoys the same regret bound and an inconsistency bound of $\mathrm{Inconsistency}(n) = O\left(\epsilon^{-2}k\log n \log(\epsilon^{-1}kn)\right)$.*

**Remark 7.1.** *The online matrix approximation with the random-order setting has previously been investigated in the context of principle component analysis by Garber et al. [2020]. They established a regret of $O\left(\zeta^{-1}\sqrt{kn}\right)$, where $\zeta$ is the smallest difference between two eigenvalues of $\mathbf{A}_t\mathbf{A}_t^\top$. In contrast, our result gives a polylogarithmic result on $\epsilon$-regret, which translate to an exact regret of $O\left(\epsilon\mathrm{OPT} + O\left(\epsilon^{-2}k\log n \log(\epsilon^{-1}kn)\right)\right)$, with $\mathrm{OPT}$ being the minimum possible cumulative loss attained by the hindsight best approximate.*

## 8 Online Regression

In the online regression problem, at each time step $t \in [n]$, we are asked to output a vector $x_t \in \mathbb{R}^d$, and then we receive vectors $a_t \in \mathbb{R}^d$ and $b_t \in \mathbb{R}$ that incurs the loss of $\ell(x_t, a_t, b_t) = \|a_t^\top x_t - b\|_2$. Without loss of generality, we assume that the losses are bounded between $[0, 1]$. We note that similar assumptions are also made in [Cesa-Bianchi et al., 1996, Ouhamma et al., 2021].

With our general reduction framework, we show an $\epsilon$-regret upper bound as follows.

**Theorem 8.1.** *For any $\epsilon \in (0,1)$, Algorithm 5 has regret $\mathrm{Regret}_\epsilon(n) = O\left(\epsilon^{-2}d\log n \log(\epsilon^{-1}dn)\right)$. Moreover, there exists an algorithm for online regression that enjoys the same regret bound and an inconsistency bound of $\mathrm{Inconsistency}(n) = O\left(\epsilon^{-2}d\log n \log(\epsilon^{-1}dn)\right)$.*

**Remark 8.1.** *In the stochastic setting, the online regression problem has been extensively investigated [Foster, 1991, Littlestone et al., 1995, Cesa-Bianchi et al., 1996, Ouhamma et al., 2021]. Using online ridge regression or forward algorithms, the regret is shown to be $O\left(d\log n\right)$. In the random-order model setting, Garber et al. [2020], Sherman et al. [2021] give $O(\sqrt{n})$-type regret when the matrix $\mathbf{A}$ has a small condition number. In comparison, our result attains polylogarithmic $\epsilon$-approximate regret, while maintaining no requirement on the loss function or the condition number. Our result can be translated to an exact regret of $O\left(\epsilon\mathrm{OPT} + O\left(\epsilon^{-2}d\log n \log(\epsilon^{-1}dn)\right)\right)$, with $\mathrm{OPT}$ being the minimum possible cumulative loss attained by the hindsight best parameter.*

## 8.1 Method and results

Similar to the low-rank matrix approximation problem, we utilize the leverage score method to learn a subspace that preserves information regarding the regression. Specifically, we use the leverage score to learn a $\epsilon$-subspace embedding, which is defined as follows.

**Definition 8.2** ($\epsilon$-Subspace Embedding). *A matrix $\mathbf{S} \in \mathbb{R}^{m \times n}$ is said to be an $\epsilon$-subspace embedding of $\mathbf{A} \in \mathbb{R}^{n \times d}$ if for any vector $x \in \mathbb{R}^d$, we have $(1-\epsilon)\|\mathbf{A}x\| \leq \|\mathbf{S}\mathbf{A}x\| \leq (1+\epsilon)\|\mathbf{A}x\|$.*

The subspace embedding serves the same functionality as coreset in the problem of online regression, it preserves the loss of information while enjoying a much lower dimension. In the online regression problem context, we define the leverage score as follows.

**Definition 8.3** (Leverage Score). *Let $\mathbf{A} = \mathbf{U}\boldsymbol{\Sigma}\mathbf{V}^\top$ be the singular value decomposition of $\mathbf{A} \in \mathbb{R}^{n \times d}$. For $i \in [n]$, the $i$-th leverage score of $\mathbf{A}$, is defined as $\tau_i = \|\mathbf{U}_{i,:}\|_2^2$.*

With the leverage score, we propose Algorithm 5. The algorithm follows the general transformation framework, where the regression problem is solved at every step with the sketch derived from the aggregated matrix using leverage score. For notational convenience, we construct the sketch by appending rows instead of columns as we did in Section 7.

---

**Algorithm 5:** Online consistent regression

**Input:** Approximation parameters $\epsilon \in (0, 1)$

1   Set $\delta = O(\epsilon/n)$ and $m = O\left(\epsilon^{-2}d\log(\delta^{-1}d)\right)$.

2   **for** $t = 1, \ldots, n$ **do**

3      Construct $\mathbf{A}_{t-1} \in \mathbb{R}^{(t-1)\times d}$ by stacking $a_1^\top, \ldots a_{t-1}^\top$.

4      Construct $b \in \mathbb{R}^{t-1}$ by stacking $b_1, \ldots, b_{t-1}$.

5      Set $\mathbf{S}^t \in \mathbb{R}^{m \times (t-1)}$ be the zero matrix.

6      **for** $j = 1, \ldots, m$ **do**

7          Sample $i \in [t-1]$ with probability $p_i := \frac{\tau_i(\mathbf{A}_{t-1})}{\sum_{j=1}^{t-1} \tau_j(\mathbf{A}_{t-1})}$.

8          Sample $w \in \mathbb{R}$ uniformly from $\left[\frac{1}{\sqrt{mp_i}}, \frac{1+\epsilon}{\sqrt{mp_i}}\right]$.

9          Replace the $j$-th row of $\mathbf{S}^t$ with $w \cdot e_i^\top$, where $e_i \in \mathbb{R}^{t-1}$ is a one-hot vector with 1 on the $i$-th index.

10      Solve the regression problem $x_t = \min_x \|\mathbf{S}^t\mathbf{A}_{t-1}x - \mathbf{S}^t b\|_2$, e.g., by an iterative method such as Newton's method.

11      Receive $a_t \in \mathbb{R}^d$, $b_t \in \mathbb{R}$, and loss $\ell(x_t, a_t, b_t)$.

---

The subspace embedding result of Woodruff [2014] immediately shows the following:

**Theorem 8.4.** *For any $\epsilon, \delta \in (0, 1)$, if $m = O\left(\epsilon^{-2}d\log(\delta^{-1}d)\right)$, then with probability $\geq 1 - \delta$, $\mathbf{S}^t$ is an $\epsilon$-subspace embedding for $\mathbf{A}_{t-1}$ with $O\left(\epsilon^{-2}d\log(\delta^{-1}d)\right)$ columns.*

To obtain Theorem 8.1, we first analyze the average sensitivity of the leverage score sampling. Then, with Theorem 8.4 and the general reduction Theorem 4.1, we obtain the regret bound.

## 9 Experiments

We here provide a preliminary empirical evaluation of our framework in the context of online $k$-means clustering, and online linear regression, with the result shown in Figure 1. Our experiments are conducted with various approximation ratios and experimental setups ($\epsilon = 0.1, 0.01, 0.001$, with $k = 3$ or $k = 5$ clusters). We then compare the performance of the proposed algorithm to the hindsight optimal solution. For $k$-means clustering, we obtain the hindsight optimal solution by applying $k$-means++ to all the data. In the context of regression, we utilize the least square formula to compute the hindsight optimal solution. Our experimental results demonstrate that the proposed algorithm is highly effective, and its performance aligns with our theoretical findings.

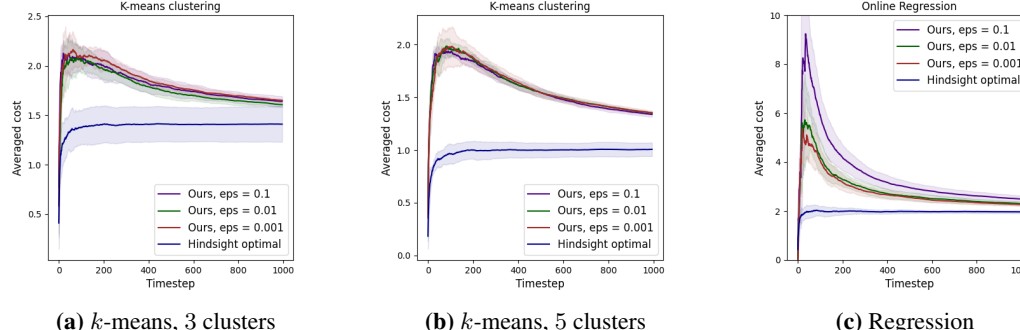

**(a)** $k$-means, 3 clusters        **(b)** $k$-means, 5 clusters        **(c)** Regression

**Figure 1:** Experimental results for $k$-means clustering with $3, 5$ clusters and online regression. Each experiment is repeated with 5 different random seed to ensure reproducible results. The shaded region indicates the 1 standard deviation.

## 10    Conclusion

In this paper, we proposed a batch-to-online transformation framework that designs consistent online approximation algorithms from offline approximation algorithms. Our framework transforms an offline approximation algorithm with low average sensitivity to an online algorithm with low approximate regret. We then show a general method that can transform any offline approximation algorithm into one with low sensitivity by using a stable coreset. To demonstrate the generality of our framework, we applied it to online $(k, z)$-clustering, online matrix approximation, and online regression. Through the transformation result, we obtain polylogarithmic approximate regret for all of the problems mentioned.

## Acknowledgement

This work is supported by JSPS KAKENHI Grant Number 20H05965 and 22H05001.

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

# A   Proofs for Section 4

**Lemma 4.2.** *Let $\mathcal{A}$ be a (randomized) algorithm for the offline learning problem with average sensitivity $\beta : \mathbb{Z}_+ \to \mathbb{R}_+$. Then for any input $X \in \mathcal{X}^n$, we have*

$$\frac{1}{n} \sum_{i=1}^{n} \mathbb{E}_{\mathcal{A}}[\ell(\mathcal{A}(X^{(i)}), x_i)] = \frac{1}{n} \sum_{i=1}^{n} \mathbb{E}_{\mathcal{A}}[\ell(\mathcal{A}(X), x_i)] \pm \beta(n),$$

*where $x = a \pm b$ means $a - b \leq x \leq a + b$.*

*Proof.* We have

$$\frac{1}{n} \sum_{i=1}^{n} \mathbb{E}_{\mathcal{A}}[\ell(\mathcal{A}(X^{(i)}), x_i)]$$

$$\leq \frac{1}{n} \sum_{i=1}^{n} \mathbb{E}_{\mathcal{A}}[\ell(\mathcal{A}(X), x_i)] + \frac{1}{n} \sum_{i=1}^{n} \left| \mathbb{E}_{\mathcal{A}}[\ell(\mathcal{A}(X), x_i)] - \mathbb{E}_{\mathcal{A}}[\ell(\mathcal{A}(X^{(i)}), x_i)] \right|$$

$$\leq \frac{1}{n} \sum_{i=1}^{n} \mathbb{E}_{\mathcal{A}}[\ell(\mathcal{A}(X), x_i)] + \frac{1}{n} \sum_{i=1}^{n} \mathrm{TV}(\mathcal{A}(X), \mathcal{A}(X^{(i)}))$$

$$\leq \frac{1}{n} \sum_{i=1}^{n} \mathbb{E}_{\mathcal{A}}[\ell(\mathcal{A}(X), x_i)] + \beta(n).$$

The other direction can be shown analogously. □

# B   Proofs for Section 5

In this section, we prove Lemma 5.2.

**Lemma B.1.** *For any $i \in [n]$ and $x \in X^{(i)}$, let $\theta^{(i)} = \mathrm{argmax}_\theta \frac{\ell(\theta,x)}{\sum_{x' \in X^{(i)}} \ell(\theta,x')}$. Then, we have*

$$0 \le \sigma_{X^{(i)}}(x) - \sigma_X(x) \le \frac{\ell\left(\theta^{(i)}, x_i\right) \cdot \ell\left(\theta^{(i)}, x\right)}{\sum_{x' \in X^{(i)}} \ell\left(\theta^{(i)}, x'\right) \cdot \sum_{x' \in X} \ell\left(\theta^{(i)}, x'\right)} \,.$$

*Proof.* Denote $\theta = \mathrm{argmax}_\theta \frac{\ell(\theta,x)}{\sum_{x' \in X} \ell(\theta,x')}$, then for the left-hand side of the inequality, we have

$$\sigma_{X^{(i)}}(x) - \sigma_X(x) \ge \frac{\ell(\theta, x)}{\sum_{x' \in X^{(i)}} \ell(\theta, x')} - \frac{\ell(\theta, x)}{\sum_{x' \in X} \ell(\theta, x')}$$

$$\ge 0 \,.$$

For the second inequality, we have

$$\sigma_{X^{(i)}}(x) - \sigma_X(x) \le \frac{\ell\left(\theta^{(i)}, x\right)}{\sum_{x' \in X^{(i)}} \ell\left(\theta^{(i)}, x'\right)} - \frac{\ell\left(\theta^{(i)}, x\right)}{\sum_{x' \in X} \ell\left(\theta^{(i)}, x'\right)}$$

$$= \frac{\ell\left(\theta^{(i)}, x_i\right) \cdot \ell\left(\theta^{(i)}, x\right)}{\sum_{x' \in X^{(i)}} \ell\left(\theta^{(i)}, x'\right) \cdot \sum_{x' \in X} \ell\left(\theta^{(i)}, x'\right)} \,.$$

$\square$

**Lemma B.2.** *For any $i \in [n]$, we have*

$$\sum_{i=1}^n \left| \sum_{x \in X} \sigma_X(x) - \sum_{x' \in X^{(i)}} \sigma_{X^{(i)}}(x') \right| \le \sum_{x \in X} \sigma_X(x) \,.$$

*Proof.* By Lemma B.1, we have

$$\sum_{x \in X} \sigma_X(x) - \sum_{x' \in X^{(i)}} \sigma_{X^{(i)}}(x') = \sigma_X(x_i) - \sum_{x \in X^{(i)}} \left( \sigma_{X^{(i)}}(x) - \sigma_X(x) \right) \le \sigma_X(x_i) \,,$$

and we have

$$\sum_{x \in X} \sigma_X(x) - \sum_{x' \in X^{(i)}} \sigma_{X^{(i)}}(x') = \sigma_X(x_i) - \sum_{x \in X^{(i)}} \left( \sigma_{X^{(i)}}(x) - \sigma_X(x) \right)$$

$$\ge \sigma_X(x_i) - \sum_{x \in X^{(i)}} \frac{\ell\left(\theta^{(i)}, x_i\right) \cdot \ell\left(\theta^{(i)}, x\right)}{\sum_{x' \in X^{(i)}} \ell\left(\theta^{(i)}, x'\right) \cdot \sum_{x' \in X} \ell\left(\theta^{(i)}, x'\right)}$$

$$= \sigma_X(x_i) - \frac{\ell\left(\theta^{(i)}, x_i\right)}{\sum_{x' \in X} \ell\left(\theta^{(i)}, x'\right)} \ge 0 \,.$$

Then, we have

$$\sum_{i=1}^n \left| \sum_{x \in X} \sigma_X(x) - \sum_{x' \in X^{(i)}} \sigma_{X^{(i)}}(x') \right| = \sum_{i=1}^n \left( \sum_{x \in X} \sigma_X(x) - \sum_{x' \in X^{(i)}} \sigma_{X^{(i)}}(x') \right)$$

$$\le \sum_{i=1}^n \sigma_X(x_i) \le \sum_{x \in X} \sigma_X(x) \,.$$

$\square$

**Lemma B.3.** *We have*

$$\sum_{i=1}^n \sum_{x \in X^{(i)}} \left| \frac{\sigma_X(x)}{\sum_{x \in X} \sigma_X(x)} - \frac{\sigma_{X^{(i)}}(x)}{\sum_{x' \in X^{(i)}} \sigma_{X^{(i)}}(x')} \right| \le 2 \,.$$

*Proof.* First, we have

$$
\frac{\sigma_X(x)}{\sum_{x\in X}\sigma_X(x)} - \frac{\sigma_{X^{(i)}}(x)}{\sum_{x'\in X^{(i)}}\sigma_{X^{(i)}}(x')}
$$
$$
= \frac{\sigma_X(x)}{\sum_{x\in X}\sigma_X(x)} - \frac{\sigma_{X^{(i)}}(x)}{\sum_{x\in X}\sigma_X(x)}\left(1 - \frac{\sum_{x'\in X^{(i)}}\sigma_{X^{(i)}}(x') - \sum_{x\in X}\sigma_X(x)}{\sum_{x'\in X^{(i)}}\sigma_{X^{(i)}}(x')}\right)
$$
$$
= \sigma_{X^{(i)}}(x)\frac{\sum_{x'\in X^{(i)}}\sigma_{X^{(i)}}(x') - \sum_{x\in X}\sigma_X(x)}{\sum_{x\in X}\sigma_X(x)\sum_{x'\in X^{(i)}}\sigma_{X^{(i)}}(x')} - \frac{1}{\sum_{x\in X}\sigma_X(x)}\left(\sigma_{X^{(i)}}(x) - \sigma_X(x)\right).
$$

We can bound this quantity from below and above by Lemma B.1,

$$
\sigma_{X^{(i)}}(x)\frac{\sum_{x'\in X^{(i)}}\sigma_{X^{(i)}}(x') - \sum_{x\in X}\sigma_X(x)}{\sum_{x\in X}\sigma_X(x)\sum_{x'\in X^{(i)}}\sigma_{X^{(i)}}(x')} - \frac{1}{\sum_{x\in X}\sigma_X(x)}\frac{\ell\left(\theta^{(i)},x_i\right)\ell\left(\theta^{(i)},x\right)}{\sum_{x'\in X}\ell\left(\theta^{(i)},x'\right)\cdot\sum_{x'\in X^{(i)}}\ell\left(\theta^{(i)},x'\right)}
$$
$$
\leq \frac{\sigma_X(x)}{\sum_{x\in X}\sigma_X(x)} - \frac{\sigma_{X^{(i)}}(x)}{\sum_{x'\in X^{(i)}}\sigma_{X^{(i)}}(x')}
$$
$$
\leq \sigma_{X^{(i)}}(x)\frac{\sum_{x'\in X^{(i)}}\sigma_{X^{(i)}}(x') - \sum_{x\in X}\sigma_X(x)}{\sum_{x\in X}\sigma_X(x)\sum_{x'\in X^{(i)}}\sigma_{X^{(i)}}(x')}.
$$

Then, we have

$$
\left|\frac{\sigma_X(x)}{\sum_{x\in X}\sigma_X(x)} - \frac{\sigma_{X^{(i)}}(x)}{\sum_{x'\in X^{(i)}}\sigma_{X^{(i)}}(x')}\right|
$$
$$
\leq \frac{\sigma_{X^{(i)}}(x)}{\sum_{x\in X}\sigma_X(x)\sum_{x'\in X^{(i)}}\sigma_{X^{(i)}}(x')}\left|\sum_{x'\in X^{(i)}}\sigma_{X^{(i)}}(x') - \sum_{x\in X}\sigma_X(x)\right|
$$
$$
+ \frac{1}{\sum_{x\in X}\sigma_X(x)}\frac{\ell\left(\theta^{(i)},x_i\right)\ell\left(\theta^{(i)},x\right)}{\sum_{x'\in X}\ell\left(\theta^{(i)},x'\right)\cdot\sum_{x'\in X^{(i)}}\ell\left(\theta^{(i)},x'\right)}.
$$

It then follows,

$$
\sum_{i=1}^{n}\sum_{x\in X^{(i)}}\left|\frac{\sigma_X(x)}{\sum_{x\in X}\sigma_X(x)} - \frac{\sigma_{X^{(i)}}(x)}{\sum_{x'\in X^{(i)}}\sigma_{X^{(i)}}(x')}\right|
$$
$$
\leq \sum_{i=1}^{n}\sum_{x\in X^{(i)}}\left(\frac{\sigma_{X^{(i)}}(x)}{\sum_{x\in X}\sigma_X(x)\sum_{x'\in X^{(i)}}\sigma_{X^{(i)}}(x')}\left|\sum_{x'\in X^{(i)}}\sigma_{X^{(i)}}(x') - \sum_{x\in X}\sigma_X(x)\right|\right.
$$
$$
\left. + \frac{1}{\sum_{x\in X}\sigma_X(x)}\frac{\ell\left(\theta^{(i)},x_i\right)\ell\left(\theta^{(i)},x\right)}{\sum_{x'\in X}\ell\left(\theta^{(i)},x'\right)\cdot\sum_{x'\in X^{(i)}}\ell\left(\theta^{(i)},x'\right)}\right)
$$
$$
= \sum_{i=1}^{n}\left(\frac{\left|\sum_{x'\in X^{(i)}}\sigma_{X^{(i)}}(x') - \sum_{x\in X}\sigma_X(x)\right|}{\sum_{x\in X}\sigma_X(x)} + \frac{1}{\sum_{x\in X}\sigma_X(x)}\frac{\ell\left(\theta^{(i)},x_i\right)}{\sum_{x'\in X}\ell\left(\theta^{(i)},x'\right)}\right)
$$
$$
\leq 1 + \frac{1}{\sum_{x\in X}\sigma_X(x)}
$$
$$
\leq 2,
$$

where the second to last inequality is from Lemma B.2, and the last inequality is by $\sum_{x\in X}\sigma_X(x) \geq 1$. □

**Lemma B.4.** *For $\epsilon > 0$, let $X$ and $X'$ be sampled from the uniform distribution over $[B,(1+\epsilon)B]$ and $[B',(1+\epsilon)B']$, respectively. Then, we have*

$$
\mathrm{TV}(X,X') \leq \frac{1+\epsilon}{\epsilon}\left|1 - \frac{B'}{B}\right|.
$$

*Proof.* The proof is implicit in Lemma 2.3 of Kumabe and Yoshida [2022]. □

**Lemma 5.2.** *The average sensitivity of Algorithm 2 is $O\left(\epsilon^{-1}m/n\right)$.*

*Proof.* The average sensitivity of importance sampling can be bounded by the sum of the average total variation distance between selected elements and that between assigned weights, conditioned on that the selected elements are the same. The former is bounded by

$$\frac{1}{n}\sum_{i=1}^{n}|C|\cdot\left(\frac{\sigma_X(x_i)}{\sum_{x\in X}\sigma_X(x)}+\sum_{x\in X^{(i)}}\left|\frac{\sigma_X(x)}{\sum_{x\in X}\sigma_X(x)}-\frac{\sigma_{X^{(i)}}(x)}{\sum_{x'\in X^{(i)}}\sigma_{X^{(i)}}(x')}\right|\right)$$
$$=O\left(\frac{|C|}{n}\right)+O\left(\frac{|C|}{n}\right)$$
$$=O\left(\frac{|C|}{n}\right),$$

where the first equality is Lemma B.3. With Lemma B.4, we have the latter as

$$\frac{1}{n}\sum_{i=1}^{n}|C|\cdot\left(\sum_{x\in X^{(i)}}\min\left\{\frac{\sigma_X(x)}{\sum_{x\in X}\sigma_X(x)},\frac{\sigma_{X^{(i)}}(x)}{\sum_{x'\in X^{(i)}}\sigma_{X^{(i)}}(x')}\right\}\cdot\frac{1+\epsilon}{\epsilon}\left|1-\frac{\sigma_{X^{(i)}}(x)/\left(\sum_{x'\in X^{(i)}}\sigma_{X^{(i)}}(x')\right)}{\sigma_X(x)/\sum_{x\in X}\sigma_X(x)}\right|\right)$$
$$\leq\frac{1}{n}\sum_{i=1}^{n}|C|\cdot\left(\sum_{x\in X^{(i)}}\frac{1+\epsilon}{\epsilon}\left|\frac{\sigma_X(x)}{\sum_{x\in X}\sigma_X(x)}-\frac{\sigma_{X^{(i)}}(x)}{\sum_{x'\in X^{(i)}}\sigma_{X^{(i)}}(x')}\right|\right)$$
$$=O\left(\frac{|C|}{\epsilon n}\right).$$

Combining the two terms, we have the average sensitivity of importance sampling be bounded as $O\left(\frac{|C|}{n}\right)+O\left(\frac{|C|}{\epsilon n}\right)=O\left(\frac{|C|}{\epsilon n}\right)$. □

## C  Proofs for Section 6

### C.1  Algorithm

We now introduced the detailed version of Algorithm 3. In Algorithm 7, we provide a detailed description of the coreset construction method for clustering, which is based on Huang and Vishnoi [2020]. To make the coreset construction enjoys small average sensitivity, we perturbe the weights assigned (Line 6 and Line 11). We show that this preserves the approximation ratio while makes the overall algorithm insensitive. With this, we obtain the online consistent clustering Algorithm 6, which clusters data from a coreset at each step.

---

**Algorithm 6:** Online consistent $(k, z)$-clustering

---

**Input:** PTAS algorithm $\mathcal{D}$ for $(k, z)$-clustering, approximation ratio $\epsilon, \delta \in (0, 1)$.

1  **for** $t = 1, \ldots, n$ **do**
2  $\quad$ Construct an $\epsilon$-coreset $C_{t-1} = (S_{t-1}, \omega_{t-1})$ by running Algorithm 7 on $X_{t-1}$.
3  $\quad$ Obtain cluster set $Z_t$ by running a PTAS $\mathcal{D}$ with approximation ratio of $(1 + \epsilon)$ on $C_{t-1}$.
4  $\quad$ Receive $x_t$ and $\ell(Z_t, x_t)$.

---

**Algorithm 7:** Coreset construction for clustering Huang and Vishnoi [2020]

---

**Input:** A set of point $X$, approximation parameter $\epsilon, \delta \in (0, 1)$, integer $k, z$.

1  Set $\epsilon = \epsilon/c$ for some large constant $c > 0$.
2  Compute a $k$-center set $C_t^* \subseteq \mathbb{R}^d$ as an $\epsilon$-approximation of the $(k, z)$-clustering problem over $X$ with the $D^z$-sampling algorithm with an approximation ratio of $O(2^z \log k)$.
3  For each $x \in X$, compute the closest point to $x$ in $C^*$, $c^*(x)$, with ties broken arbitrarily. For each $c \in C^*$, denote $X^c$ to be the set of points $x \in X$ with $c^*(x) = c$.
4  For each $x \in X$, let $\sigma_{1,X}(x) = 2^{2z+2}\epsilon^2 \left( \frac{\|x - c^*(x)\|_2^z}{\sum_{x \in X} \ell(C^*, x)} + \frac{1}{|X_{c^*(x)}|} \right)$.
5  Pick a non-uniform random sample $D^1$ of $N_1 = O\left((168z)^{10z}\epsilon^{-5z-15}k^5 \log \frac{k}{\delta}\right)$ points from $X$, where each $x \in X$ is selected with probability $\frac{\sigma_{1,X}(x)}{\sum_{y \in X} \sigma_{1,X}(y)}$.
6  For each $x \in D^1$, sample $\tilde{u}_X(x)$ from $\left[ \frac{\sum_{y \in X} \sigma_{1,X}(y)}{|D_t^1| \cdot \sigma_{1,X}(x)}, (1 + \epsilon) \frac{\sum_{y \in X} \sigma_{1,X}(y)}{|D_t^1| \cdot \sigma_{1,X}(x)} \right]$.
7  Set $u_X(x) = \tilde{u}_X(x)$
8  For each $c \in C^*$, compute $D_c$ to be the set of points in $D^1$ whose closest point in $C^*$ is $c$ with ties broken arbitrarily.
9  For each $x \in D^1$, let $\sigma_{2,X}(x) = \frac{u_X(x) \cdot \ell(C^*, x)}{\sum_{y \in D^1} u_X(y) \cdot \ell(C^*, y)}$
10  Pick a non-uniform random sample $D^2$ of $N_2 = O\left(\epsilon^{-2z-2}k \log k \log \frac{k}{\epsilon\delta}\right)$ points from $X_t$, where each $x \in X_t$ is selected with probability $\frac{\sigma_{2,X}(x)}{\sum_{y \in D^1} \sigma_{2,X}(y)}$
11  For each $x \in D^2$, sample $\tilde{w}_X(x)$ from $\left[ \frac{\sum_{y \in D^1} \sigma_{2,X}(y)}{|D^2| \cdot \sigma_{2,X}(x)}, (1 + \epsilon) \frac{\sum_{y \in D^1} \sigma_{2,X}(y)}{|D^2| \cdot \sigma_{2,X}(x)} \right]$.
12  Set $w_X(x) = \tilde{w}_X(x)$
13  For each $c \in C^*$, let $w_X(c) = (1 + 10\epsilon) \sum_{x \in D_c} u_X(x) - \sum_{x \in D^2 \cap D_c} w_X(x)$.
14  $S = D^2 \cup C^*$
15  $w(x) = w_X(x)$
16  Output $(S, w)$

---

### C.2  Analysis

**Theorem C.1.** *Algorithm 7 outputs an $\epsilon$-coreset with probability at least $1 - \delta$ and has an average sensitivity of*

$$O\left( \frac{k + (168z)^{10z}\epsilon^{-5z-15}k^6 \log \frac{k}{\delta} + \epsilon^{-2z-2}k \log k \log \frac{k}{\epsilon\delta}(1 + \epsilon^{-2z-2}k^2 \log k \log \frac{k}{\epsilon\delta})}{n} \right).$$

*Proof.* We first show that our coreset construction method gives an $\epsilon$-coreset.

We remark that the way we assigned the weights is a perturbed version of the method presented in Huang and Vishnoi [2020]. Then, to show that the perturbed version of the assignment of the weight still preserves the coreset approximation ratio, we show that the perturbed version still satisfies Corollary 15.3 of Feldman and Langberg [2011] and Theorem 14.5. Then using the same argument as Theorem 15.4 of Feldman and Langberg [2011], our coreset construction method gives an $\epsilon$-coreset.

In the first importance sampling stage, we only perturb $\tilde{u}_X(x)$ by a ratio of $1 + \epsilon$, for each $x \in D^1$. The same is applied in the second stage with $\tilde{w}_X(x)$, for all $x \in D^2$. For $c \in C^*$, we have

$$(1 + 10\epsilon) \sum_{x \in D_c} u_X(x) - \sum_{x \in D^2 \cap D_c} w_X(x) \le (1 + \epsilon) \left( (1 + 10\epsilon) \sum_{x \in D_c} \tilde{u}_X(x) - \sum_{x \in D^2 \cap D_c} \tilde{w}_X(x) \right) .$$

In all cases, the result from [Feldman and Langberg, 2011] still holds, as the weights are scaled by $(1 + \epsilon)$ at most. By the same argument that all perturbed weights are scaled by $(1 + \epsilon)$ at most, we have Theorem 15.4 of Feldman and Langberg [2011] holds with a ratio of $\epsilon(1 + \epsilon)$. This results in an approximation of $1 + \epsilon(1 + \epsilon)$ by applying the argument of Theorem 15.4 of Feldman and Langberg [2011]. Rescale $\epsilon$ to $\epsilon/c$ for some constant $c > 0$ gives an approximation ratio of $1 + \epsilon$ and completes our argument that Algorithm 7 produces an $\epsilon$-coreset with probability at least $1 - \delta$.

To show that our method enjoys low average sensitivity, we upper bound the average sensitivity of the two importance sampling stage of Algorithm 7 separately.

The average sensitivity of importance sampling in the first stage can be bounded by the average total variation distance between selected elements and that between assigned weights, conditioned on the selected elements are the same. By Lemma 5.2, we can upper bound the average sensitivity of the first importance sampling stage to be $O(N_1/n)$, where $N_1 = O\left( (168z)^{10z} \epsilon^{-5z-15} k^5 \log(\delta^{-1}k) \right)$.

In the second stage, as points are selected with probability $\frac{\sigma_{2,X}(x)}{\sum_{y \in X} \sigma_{2,X}(y)}$, we bound the total variation distance between selected points in a similar way as to that for the first stage. This gives a bound of $O(N_2/n)$, where $N_2 = O\left( \epsilon^{-2z-2} k \log k \log \frac{k}{\epsilon\delta} \right)$. To bound the average total variation distance between assigned weights, we again apply the same argument as above and obtain a bound of $O(N_2/n)$.

Combining these and that the average sensitivity of Line 2 of Algorithm 7 is $O(k/n)$ (by Lemma 2.2 of Yoshida and Ito [2022]), we have the average sensitivity as

$$O\left( \frac{k + N_1 + N_2}{n} \right) = O\left( \frac{k + (168z)^{10z} \epsilon^{-5z-15} k^5 \log \frac{k}{\delta} + \epsilon^{-2z-2} k \log k \log \frac{k}{\epsilon\delta}}{n} \right) . \qquad \square$$

**Theorem 6.1.** *For any $\epsilon \in (0, 1)$, Algorithm 3 gives a regret bound of*

$$\text{Regret}_\epsilon(n) \le O\left( \left( (168z)^{10z} \epsilon^{-5z-15} k^5 \log \frac{kn}{\epsilon} + \epsilon^{-2z-2} k \log k \log \frac{kn}{\epsilon} \right) \log n \right) .$$

*Moreover, there exists an algorithm that enjoys the same regret bound and an inconsistency bound of* $\text{Inconsistency}(n) = O\left( \left( (168z)^{10z} \epsilon^{-5z-15} k^5 \log(\epsilon^{-1}kn) + \epsilon^{-2z-2} k \log k \log(\epsilon^{-1}kn) \right) \log n \right)$ *for $(k, z)$-clustering.*

*Proof.* First, we note that the approximation ratio of the algorithm with respect to the aggregated loss is at most

$$(1 - \delta) \left( 1 + \frac{\epsilon}{3} \right) \left( 1 + \frac{\epsilon}{3} \right) + \delta n \le 1 + \epsilon$$

from the choice of $\delta$, i.e., $\delta = O(\epsilon/n)$.

Also, we note that the overall average sensitivity is

$$\sum_{t=1}^{n} \beta(t) = \sum_{t=1}^{n} O\left( \frac{(168z)^{10z} \epsilon^{-5z-15} k^5 \log \frac{k}{\delta} + \epsilon^{-2z-2} k \log k \log \frac{k}{\epsilon\delta}}{t} \right)$$

$$= O\left( \left( (168z)^{10z} \epsilon^{-5z-15} k^5 \log \frac{k}{\delta} + \epsilon^{-2z-2} k \log k \log \frac{k}{\epsilon\delta} \right) \log n \right) .$$

Then substituting this into Theorem 4.1, for $z \geq 1$, we have a regret bound of

$$\text{Regret}_\epsilon(n) \leq O\left(\left((168z)^{10z}\epsilon^{-5z-15}k^5 \log\frac{k}{\delta} + \epsilon^{-2z-2}k \log k \log\frac{k}{\epsilon\delta}\right)\log n\right)$$

$$= O\left(\left((168z)^{10z}\epsilon^{-5z-15}k^5 \log\frac{kn}{\epsilon} + \epsilon^{-2z-2}k \log k \log\frac{kn}{\epsilon}\right)\log n\right).$$

For the second claim about inconsistency, we first convert the average sensitivity bound to an inconsistency bound of the same order. This can be done by Lemma 4.2 of [Yoshida and Ito, 2022] and by arguing in a similar way as Lemma 4.5 of [Yoshida and Ito, 2022] by showing that there exists a computable probability transportation for the output of Algorithm 3. This thus yields an inconsistency of $O\left(\left((168z)^{10z}\epsilon^{-5z-15}k^5 \log(\epsilon^{-1}kn)\epsilon^{-2z-2}k \log k \log(\epsilon^{-1}kn)\right)\log n\right)$. $\qquad\square$

## D   Proofs for Section 7

**Theorem 7.2.** *For any $\epsilon \in (0,1)$, Algorithm 4 has regret $\mathrm{Regret}_\epsilon(n) = O\left(\epsilon^{-2}k \log n \log(\epsilon^{-1}kn)\right)$. Moreover, there exists an algorithm for online low-rank matrix approximation that enjoys the same regret bound and an inconsistency bound of $\mathrm{Inconsistency}(n) = O\left(\epsilon^{-2}k \log n \log(\epsilon^{-1}kn)\right)$.*

*Proof.* Let $\delta > 0$ be determined later. By Theorem 6 from Cohen et al. [2017], with probability $1 - \delta$, for any rank-$k$ orthogonal projection $\mathbf{X}$, by sampling $m = O\left(\epsilon^{-2}k \log(\delta^{-1}k)\right)$ columns, in Line 4 of Algorithm 5, we have,

$$\left(1 - \frac{\epsilon}{2}\right)\|\mathbf{A}_t - \mathbf{X}\mathbf{A}_t\|_F^2 \le \|\mathbf{C}_t - \mathbf{X}\mathbf{C}_t\|_F^2 \le \left(1 + \frac{\epsilon}{2}\right)\|\mathbf{A}_t - \mathbf{X}\mathbf{A}_t\|_F^2.$$

With this, we note that the algorithm has an approximation of $1 + \epsilon$ as $1 + \epsilon/2 + \delta n \le 1 + \epsilon$ by choosing the hidden constant in $\delta$ small enough.

Note that by applying Lemma 5.2, this routine has an average sensitivity of $O(m/t) = O\left(\epsilon^{-2}k \log(\delta^{-1}k)/t\right)$ at any step $t$. Then set $\mathbf{Z}_t$ to the top $k$ left singular vectors of $\mathbf{C}_t$, and we have

$$\left\|\mathbf{A}_t - \mathbf{Z}_t\mathbf{Z}_t^\top \mathbf{A}_t\right\|_F \le (1 + \epsilon)\left\|\mathbf{A}_t - \mathbf{U}_k\mathbf{U}_k^\top \mathbf{A}_t\right\|_F.$$

To obtain the regret, we calculate the overall average sensitivity as $\sum_{t=1}^n \beta(t) = \sum_{t=1}^n O\left(\epsilon^{-2}k \log(\delta^{-1}k)/t\right) = O\left(\epsilon^{-2}k \log n \log(\delta^{-1}k)\right)$. Applying Theorem 4.1, and from the choice $\delta = O(\epsilon/n)$, we have $\mathrm{Regret}_\epsilon(n) = O\left(\epsilon^{-2}k \log n \log(\epsilon^{-1}kn)\right)$.

For the second claim about inconsistency, we prove this convert the average sensitivity bound to an inconsistency bound of the same order. This can be done by Lemma 4.2 of [Yoshida and Ito, 2022] and by arguing in a similar way as Lemma 4.5 of [Yoshida and Ito, 2022] by showing that there exists a computable probability transportation for the output of Algorithm 3.   $\square$

# E   Proof for Section 8

**Theorem 8.1.** *For any $\epsilon \in (0, 1)$, Algorithm 5 has regret $\text{Regret}_\epsilon(n) = O\left(\epsilon^{-2} d \log n \log(\epsilon^{-1} dn)\right)$. Moreover, there exists an algorithm for online regression that enjoys the same regret bound and an inconsistency bound of* $\text{Inconsistency}(n) = O\left(\epsilon^{-2} d \log n \log(\epsilon^{-1} dn)\right)$.

*Proof.* By Theorem 8.4, the algorithm has an approximation of $1 + \epsilon$ as $1 + \epsilon/2 + \delta n \leq 1 + \epsilon$ by choosing the hidden constant in $\delta$ small enough.

Similar to the low-rank case, the average sensitivity of leverage score sampling at step $t$ is $O\left(\frac{d \log(d/\delta)}{\epsilon^2 t}\right)$. Summing over the average sensitivity

$$\sum_{t=1}^{n} O\left(\frac{d \log(d/\delta)}{\epsilon^2 t}\right) = O\left(\frac{d \log n \log(d/\delta)}{\epsilon^2}\right).$$

Take this into Theorem 1, and with $\delta = O(\epsilon/n)$, we have a $\epsilon$-regret of $O\left(\epsilon^{-2} d \log n \log(dn/\epsilon)\right)$.

Similar to that of Theorem 6.1 and Theorem 7.2, we prove the second claim by converting the average sensitivity bound to an inconsistency bound of the same order. This can be done by Lemma 4.2 of [Yoshida and Ito, 2022] and by arguing in a similar way as Lemma 4.5 of [Yoshida and Ito, 2022] by showing that there exists a computable probability transportation for the output of Algorithm 3. □

