# OpenReview forum: "A Batch-to-Online Transformation under Random-Order Model"
_NeurIPS.cc/2023/Conference — NeurIPS 2023 poster_

### Official Review · Reviewer_zXrL · 2023-07-03

**Soundness:** 4 excellent
**Presentation:** 3 good
**Contribution:** 2 fair
**Rating:** 6
**Confidence:** 3

**Summary:**

This is a paper about online learning in the random arrival/order model, i.e., the adversary chooses samples but they are randomly permuted before being presented to an online algorithm. The authors propose a general tool that establishes a reduction to the offline/batch setting. The bounds that one can obtain with that tool are in terms of approximate regret, i.e., on top of the standard regret one also loses an epsilon fraction of the optimum.  The key idea behind the reduction is to use a coreset to reduce the sensitivity of the offline algorithm; the rest is fairly straightforward. After presenting the general tools, the authors show how to apply it to three specific problems: clustering, low-rank matrix approximation, regression.

**Strengths:**

The main result is a general tool that potentially can be applied to different problems.

Obtaining this result required a certain level of technical mastery, and was not completely trivial.

The paper is well written and easy to read.

**Weaknesses:**

Technical novelty is very limited. The paper combines known results in a very natural (though technically difficult) way. One could argue that the hard (and novel) part of the job was choosing the right pieces to put them together – but e.g. the idea of using coresets in a very similar context appears already e.g. in [Cohen-Addad et al., 2021] (the authors do not mention that).

There are no experiments – even though one of the applications is a clustering problem studied by Cohen-Addad et al. [2021], who provided some preliminary experimental results, so it would be fairly easy to compare to them. It would be particularly interesting because the theoretical bounds in the two papers are not directly comparable.

Even though the paper provides new algorithms for three problems previously studied in the literature, because of the specific setup, the theoretical bounds are, as far as I understand, incomparable to the previous ones.

The results allow only for approximate regret bounds (and only under random arrivals). It seems to me to be a big limitation, and the authors did not manage to convince me that this is not the case.

**Questions:**

I’d be curious to see an experimental comparison with [Cohen-Addad et al., 2021].

Definition 3.1 (approximate regret) seems to be a standard one, so you could provide a reference and a bit more of a context.

Minor remarks:

Line 116: Is F supposed to be A?

Lines 123, 235: \citet -> \citep

Line 166: data -> data point

Line 191: logarithmic -> polylogarithmic

Algorithm 3, Input: “approximation ratio eps \in (0, 1)” -> “approximation ratio 1+eps, eps \in (0, 1)”

Line 221: square -> squares

Lines 227, 259: assumptions are also assumed -> assumptions are also made

Line 253: outermost brackets should be bigger, there should not be one O inside of another

Line 270: missing closing bracket

Line 276: a -> an

Line 288: \citep -> \citet

Line 295: “an offline approximation algorithm” -> “offline approximation algorithms”

Line 364: the second to last inequality requires some justification

Line 395: insensitivity -> insensitive, algorithm 6 -> Algorithm 6

**Limitations:**

The results yield only approximate regret bounds, which seems to me somewhat limiting.

---

> ### Author Rebuttal · Authors · 2023-08-08
>
> We thank the reviewer for the insightful comments. In the following, we address the raised concerns point by point. Minor comments and typos have already been addressed and corrected in our manuscript.
>
> # 1. Technical Novelty
> We respectfully **disagree** with the reviewer's comment suggesting that our results are merely a combination of known results. The example from [Cohen-Addad et al., 2021], which uses coresets for $k$-means clustering, does not diminish the originality and novelty of our approach. It is important to note that while both works employ coresets, they serve different purposes in entirely distinct contexts. In [Cohen-Addad et al., 2021], coresets are used to reduce the number of experts in the MWUA algorithm, whereas, in our work, coresets are used to achieve a small average sensitivity.
>
> Our main technical contribution is to the general framework for batch-to-online transformation via the notion of average sensitivity, which applies to a range of applications. Our results for $(k,z)$-clustering is only an example of applications of the general transformation framework.
>
> On the $(k,z)$-clustering problem, we also want to emphasize that the scope of our results extends beyond the $k$-means clustering discussed in [Cohen-Addad et al., 2021]. Our findings are applicable to the general $(k, z)$ clustering problem, where $k$-means represents a specific case for $z = 2$. It is also worth noting that using a dimension-independent coreset method for general clustering with the algorithm proposed by [Cohen-Addad et al., 2021] does not automatically lead to sublinear and dimension-independent regret. This is primarily because other components of their algorithm still rely on the assumption that $z = 2$, and they do not permit dimension-independent results.
>
> # 2. Experiments
> We first would like to remark that Cohen-Addad et al.~[2021] only provided an example of FTL and FLT-MWUA in $k$-means clustering to demonstrate their poor performance. The proposed algorithm (Algorithm 1, $\epsilon$-regret minimization) is not empirically evaluated. This, as well as the absence of released example code, prevents us from conducting a direct comparison to their approach.
>
> It is also important to highlight that our work is primarily theoretical in nature, as our focus lies in exploring and analyzing the theoretical foundations of the general framework proposed. Nonetheless, we provide a preliminary empirical evaluation of our framework in the context of online $k$-means clustering, online linear regression (see the additional file in the general rebuttal), and with various approximation ratios and experimental setup ($\epsilon = 0.1, 0.01, 0.001$, with $k=3$ or $k=5$ clusters). We compare the performance of the proposed algorithm to the hindsight optimal solution. For $k$-means clustering, we obtain the hindsight optimal solution by applying $k$-means++ to all the data. In the context of regression, we utilize the least square formula to compute the hindsight optimal solution. Our experimental results demonstrate that the proposed algorithm is highly effective, and its performance aligns with our theoretical findings.
>
> # 3. Exact regret bound and comparison to previous bounds
> In the case where the problem is NP-complete (for example, $(k,z)$-clustering), no efficient algorithm can attain sublinear exact regret, and thus our framework can only obtain approximate regret.
>
> Using the definition of approximate regret, we can rearrange the terms and obtain
> $$\mathbb{E} _{\mathcal{A},\\{x _t \\}}\left[\sum^n _{t=1} \ell(\theta _t, x _t) - \mathrm{OPT}\right]  = \epsilon  \mathrm{OPT} + \mathrm{Regret} _\epsilon(n) .$$
> Then by applying the (generalized) AM-GM inequality to the right-hand side and tuning $\epsilon$ appropriately so the inequality holds as equality,  we can obtain an exact regret.
>
> For example, from the approximate regret of our online matrix approximation result, we can translate it to $O(\mathrm{OPT}^{2/3} (d \log n \log(\epsilon^{-1} kn))^{1/3})$. From the approximate regret of our online regression result, we can translate it to $O(\mathrm{OPT}^{2/3} (d \log n \log(\epsilon^{-1} dn))^{1/3})$.
>
> We also want to highlight that, in the case of $k$-means ($z = 2$), our Theorem 5.1 gives a logarithmic regret bound (on $n$), which is also dimension independent. In comparison [Cohen-Addad et al., 2021] depend on the dimension $d$ and give a $O(\sqrt{n})$ for regret. For online regression, [Garber et al., 2020] give $O(\sqrt{n})$-type regret when the matrix $\boldsymbol{A}$ has a small condition number. In comparison, our result attains polylogarithmic $\epsilon$-approximate regret, while maintaining no requirement on the loss function or the condition number.
>
> # 4. Line 364, Lemma 4.2, second to last inequality
>  Let $\mathcal{D}$ be the distribution over pairs of outputs that attains $\mathrm{TV}(\mathcal{A}(X), \mathcal{A}(X^{(i)}))$.
>     Then, we have
> $$\mathbb{E }_{\mathcal{A}}[\ell(\mathcal{A}(X),x_i)] - \mathbb{E} _{\mathcal{A}} [\ell(\mathcal{A}(X^{(i)}), x_i)] =
>  \mathbb{E} _{(\theta,\theta^{(i)}) \sim \mathcal{D}}[\ell(\theta,x_i) - \ell(\theta^{(i)}, x_i)] $$
> $$\leq \mathrm{TV}(\mathcal{A}(X), \mathcal{A}(X^{(i)})) \cdot \max _{\theta'} \ell(\theta', x_i)
>         \leq \mathrm{TV}(\mathcal{A}(X), \mathcal{A}(X^{(i)})) , $$
> where the last inequality is by the assumption that $\ell(\cdot,\cdot) \leq 1$ (In the case where this is not true, we can normalize the losses to satisfy this assumption).

---

> > ### Comment · Reviewer_zXrL · 2023-08-11
> >
> > Thank you for the response. Could you please elaborate a bit more on point 3? How exactly do you apply AM-GM and tune epsilon to get such a bound for online matrix approximation?

---

> > > ### Author Response · Authors · 2023-08-11
> > > **Using AM-GM to get exact regret bound**
> > >
> > > From the definition of regret, we can obtain
> > > $\mathbb{E} _{\mathcal{A},\{x _t \}}\left[\sum^n _{t=1} \ell(\theta _t, x _t) - \mathrm{OPT}\right]  = \epsilon  \mathrm{OPT} + \mathrm{Regret} _\epsilon(n) $. In the case of online matrix approximation and online regression, $ \mathrm{Regret} _\epsilon(n) = O \left(\epsilon^{-2}k \log n\log(\epsilon^{-1}kn)\right)$ and $\mathrm{Regret} _\epsilon(n) = O\left(\epsilon^{-2}d \log n \log (\epsilon^{-1}dn)\right)$ respectively. Let us short hand this term to be $\mathrm{Regret} _\epsilon(n) = O (B / \epsilon^{-2})$.
> > >
> > > We now use a generalized AM-GM inequality, $\epsilon  \mathrm{OPT} + B / \epsilon^{-2} \geq \left(  \epsilon^2  \mathrm{OPT}^2 \cdot B/\epsilon^2 \right)^{1/3}$. We then pick $\epsilon = O \left( \frac{B ^{1/3}}{\mathrm{OPT}^{1/3}} \right)$ to make it holds as equality. Take this value of $\epsilon$ in gives a regret bound of $O \left( B^{1/3}  \mathrm{OPT}^{2/3}\right)$, which is $O(\mathrm{OPT}^{2/3} (k \log n \log(\epsilon^{-1} kn))^{1/3})$ in the case of matrix approximation.

---

> > > > ### Comment · Reviewer_zXrL · 2023-08-11
> > > >
> > > > Thank you for a quick reply. Given all the above explanations of how your contributions fit the context of prior work, I increase my initial evaluation to 6.

---

> > > > > ### Author Response · Authors · 2023-08-11
> > > > > **Thank you!**
> > > > >
> > > > > We thank the reviewer for the positive feedback!

---

### Official Review · Reviewer_uJY4 · 2023-07-04

**Soundness:** 4 excellent
**Presentation:** 3 good
**Contribution:** 3 good
**Rating:** 8
**Confidence:** 3

**Summary:**

This paper provides a general framework for how to use good offline methods to construct online methods under the random-order setting.  It points out the importance of the average sensitivity in the transformation from good offline methods to good online methods.

**Strengths:**

1. The framework is very general for the algorithm design of online algorithms based on offline algorithms.

2. The connection between regret and average sensitivity is beautifully characterized.

**Weaknesses:**

1. It looks the method cannot cover the setting with $\epsilon=0$. If you can, please clarify it and I will further raise my rating.

2. it is also better to have some numerical experiments of the performance on some specific problems.

**Questions:**

1. can this method also solve the case with $\epsilon=0$? At least for the online linear programming problem, offline LPs can be accurately solved, and the regret (epsilon=0) can also be small under the random-order setting. Can this setting also be converted by your paper?

---

> ### Author Rebuttal · Authors · 2023-08-08
>
> We thank the reviewer for the insightful and positive comments! In the following, we address the raised concerns point by point.
>
> # 1. Recover $\epsilon = 0$ (exact regret)
>
> We can obtain exact regret for our algorithms. Using the definition of approximate regret, we can rearrange the terms and obtain $\mathbb{E} _{\mathcal{A},\\{x_t\\}}\left[\sum^n _{t=1} \ell(\theta_t, x_t) - \mathrm{OPT}\right] = \epsilon  \mathrm{OPT} + \mathrm{Regret} _\epsilon(n) $. Then by applying the (generalized) AM-GM inequality to the right-hand side and tuning $\epsilon$ appropriately so the inequality holds as equality, we can obtain an exact regret.
> From the approximate regret of our online matrix approximation result, we can translate it to $O(\mathrm{OPT}^{2/3} (d \log n \log(\epsilon^{-1} kn))^{1/3}) $.
> From the approximate regret of our online regression result, we can translate it to $O(\mathrm{OPT}^{2/3} (d \log n \log(\epsilon^{-1} dn))^{1/3})$.
>
> # 2. Online LP
> We believe that our general transformation framework can provide insights into new algorithms for online LPs in the random-order setting. Let us consider the following online integer LP (the binary version of this is also studied in [Li et al., 2020],
> $ \max \ r^{\top} x , \text { s.t. } A x \leq b$, where $r =\left(r_1, \ldots, r_n\right)^{\top} \in \mathbb{R}^n, \boldsymbol{A}=\left(a_1, \ldots, a_n\right) \in \mathbb{R}^{m \times n}$, and $b =\left(b_1, \ldots, b_m\right)^{\top} \in \mathbb{R}^m$. Here $a_j=\left(a_{1 j}, \ldots, a_{m j}\right)^{\top}$ denotes the $j$-th column of the constraint matrix $\boldsymbol{A}$. At each time step $t$, we receive $r_t, a_t$ and are asked to compute $x_t$. In the offline case, this can be exactly solved by the cutting plane or ellipsoid methods. In the online case, we first want to choose a subset of columns and weights,
>
> $A=(a'_i,\ldots,a'_k), w_1,\ldots,w_k$
>
> such that $Ax \leq b$ (approximately) holds whenever $\sum_{i=1}^k w_i a'_i x_i \leq b$ holds. This step is similar to lines $6$-$9$ of Algorithm 5. Then one can use an exact method at each step on the sketched matrix. Using a similar analysis of Theorem 8.1, one should be able to obtain a regret bound for online linear programming. We leave the detailed derivation of the algorithm and analysis for future work.
>
> Li, X., Sun, C., \& Ye, Y. (2020). Simple and fast algorithm for binary integer and online linear programming. Advances in Neural Information Processing Systems, 33, 9412-9421.
>
> # 2.  Experiments
>
> We would first like to highlight that our work is primarily theoretical in nature, as our focus lies in exploring and analyzing the theoretical foundations of the general framework proposed. Moreover, the related works are mostly theoretical as well, such as [Cohen-Addad et al., 2021] (for online clustering), [Garber et al., 2020] (for online regression, and online matrix approximation in PCA). Due to the scarcity of available code for baseline algorithms, it is hard to conduct a fair empirical evaluation of our approach in the limited time of the author's response period.
>
> Nonetheless, we here provide a preliminary empirical evaluation of our framework in the context of online $k$-means clustering, online linear regression (see the additional file in the overall rebuttal file), and various approximation ratios and experimental setup ($\epsilon = 0.1, 0.01, 0.001$, with $k=3$ or $k=5$ clusters). We compare the performance of the proposed algorithm to the hindsight optimal solution. For $k$-means clustering, we obtain the hindsight optimal solution by applying $k$-means++ to all the data. In the context of regression, we utilize the least square formula to compute the hindsight optimal solution. Our experimental results demonstrate that the proposed algorithm is highly effective, and its performance aligns with our theoretical findings.

---

> > ### Comment · Reviewer_uJY4 · 2023-08-13
> >
> > Thank you for these clarifications and answers. I raise my rating, especially for the answer to the first question.

---

> > > ### Author Response · Authors · 2023-08-13
> > >
> > > Thank you for championing our paper!

---

### Official Review · Reviewer_4met · 2023-07-05

**Soundness:** 3 good
**Presentation:** 3 good
**Contribution:** 3 good
**Rating:** 7
**Confidence:** 2

**Summary:**

The paper presents a general method for converting an offline approximation to a learning problem into an online algorithm for the random-order problem which enjoys low (epsilon-approximate) regret. Specifically, the regret of this method depends on the sensitivity of the offline approximation; intuitively, this is the amount by which the distribution of outputs for the algorithm changes upon a small change to the input.

First, the paper presents the construction for converting offline approximation algorithms to online algorithms for the random-order model. This construction is quite simple: at any step, we run the offline approximation on the part of the input that has already arrived, and use the output as the choice of the online algorithm for the next step. This is shown to yield low epsilon-approximate regret when the offline algorithm has low sensitivity.

Next, the paper presents/analyzes low-sensitivity offline algorithms for a few online problems (online clustering, online matrix approximation and online regression); these algorithms can then be plugged into the aforementioned framework in order to yield regret bounds for these online problems.

To my understanding, the main algorithmic concept used in designing low-sensitivity offline approximation is using coresets, which are subsets of the input that accurately represent the overall input. The paper analyzes a method for constructing such coresets, and shows that it has low-sensitivity. This coreset construction is then used to choose stable subsets of the input, yielding online algorithms with low regret and low inconsistency (which is a measure of changes in the online algorithm's output over time).

**Strengths:**

The algorithmic framework introduced is general, and could be applied to additional online learning problems.
The paper introduces the goal of having low-sensitivity offline approximations for the sake of this offline-to-online conversion; this could be used to guide future research efforts.
The algorithmic technique used for designing low-sensitivity algorithms in the paper is nice.


**Weaknesses:**

The offline-to-online conversion itself is quite simple, and does not introduce any algorithmic techniques (arguably, this is the first conversion that comes to mind).


**Questions:**

none

**Limitations:**

yes

---

> ### Author Rebuttal · Authors · 2023-08-08
>
> We thank the reviewer for the positive comments! We hope to address the concerns raised in the following response.
>
> > The offline-to-online conversion itself is quite simple and does not introduce any algorithmic techniques (arguably, this is the first conversion that comes to mind).
>
> We believe that the online-to-offline reduction through low average sensitivity, while may be seemingly straightforward, can provide insights into developing new online learning algorithms. Specifically, the general transformation framework provides a simple recipe for designing low-regret algorithms for a range of online learning problems with off-the-shelf offline algorithms.
>
> We would also like to note that although the overall conversion might seem straightforward, much more algorithmic techniques and insights are required to design appropriate coreset construction methods and obtain the theoretical guarantee. For example, in order to obtain a low average sensitivity coreset construction method, Algorithm 2 is proposed with a careful perturbation on the weights of coresets. The specific algorithm used for deriving the results for the applications, such as clustering, requires even more design techniques to obtain such dimension-independent results.

---

> > ### Comment · Reviewer_4met · 2023-08-15
> >
> > Thanks for your response.

---

### Official Review · Reviewer_A1nn · 2023-07-14

**Soundness:** 4 excellent
**Presentation:** 4 excellent
**Contribution:** 3 good
**Rating:** 6
**Confidence:** 4

**Summary:**

The authors present a framework for reducing online to offline learning. More presicely, they consider random order online learning where at each round a random point of some domain $X$ is presented to the learner (as opposed to some adversary choosing the example). They first show that an offline algorithm with low average sensitivity, i.e., the output of the algorithm does not depend too much on any single point, then it can be effectively used to solve the corresponding online problem via a standard "follow-the-leader" type reduction. In particular, at round $t$ having observed the points $X_{t-1} = x_1,\ldots, x_{t-1}$ in the
online learning setting, the learner runs the offline algorithm on $X_{t-1}$ and uses its answer to play at the current round. They next show how to obtain offline algorithms with low average sensitivity by constructing a coreset via sensitivity sampling.  They apply their framework to various problems such as online clustering and online linear regression.


**Strengths:**


The online-to-batch problem considered in this work is well-motivated and interesting. The authors provide a unified approach to obtain online algorithms for random-order online learning tasks and show some non-trivial results for popular online problems such as online regression.  I believe that those results are going to be of interest for the NeurIPS community.

The applications of the framework proposed in this work yield improved regret bounds compared to the prior work.  In the case of online matrix approximation application, this work provides a regret bound that is only logarithmic in the horizon $n$ while the prior work the regret was roughly $\sqrt{n}$ and also depended on a ``condition number'' of the observed matrices.

The paper is very carefully written and the main results and proofs were easy to follow.


**Weaknesses:**

This work considers the weaker online learning model of random order (instead of the more standard adversarial model).  In Remark 6.1 (online clustering)
the authors compare with online algorithms that dealt with the adversarial setting.  I think a table with precise comparisons with prior work would help this paper.

The online to offline reduction in the random ordering assuming low average sensitivity is not very surprising and the proof (see proof of Lemma 4.2) is rather straightforward. Also, it seems to crucially rely on the fact that the online learning is random order. Would a this or a similar approach based on low average sensitivity yield any guarantee for online learning in the adversarial setting?


**Questions:**

See weaknesses.

**Limitations:**

Yes

---

> ### Author Rebuttal · Authors · 2023-08-08
>
> We thank the reviewer for the insightful comments!
>
> # Applicability in adversarial setting and comparison to the previous bounds
> We thank the reviewer for the constructive advice, we added a comparison table for the results we obtained in our manuscript.
>
> We believe that the online-to-offline reduction through low average sensitivity, while may be seemingly straightforward, can provide insights into developing new online learning algorithms. Specifically, the general transformation framework provides a recipe for designing low-regret algorithms for a range of online learning problems with off-the-shelf offline algorithms. We also want to remark that in section 5, we show that the low average sensitivity assumption can be realized via insensitive coreset construction methods.
>
> Regarding the applicability of our framework in the adversarial setting, we do not think the argument applies to the adversarial setting, as $\sum_{t} \ell(A(X_t), x_{t+1}) - \ell(A(X_{t+1}), x_{t+1}) \leq \sum_t \beta(t)$ (Lemma 4.2) does not hold in the adversarial setting. This is because we can only guarantee that on average, removing a point does not change $X_{t+1}$. However, we cannot guarantee the worst case. So for every $t$, removing $x_{t+1}$ might be the worst change to $X_{t+1}$.

---

> > ### Comment · Reviewer_A1nn · 2023-08-17
> > **Thanks for the response**
> >
> > I would like to thank the authors for their response. I remain in favor of acceptance of this work.

---

> > > ### Author Response · Authors · 2023-08-17
> > >
> > > Thank you for championing our work!

---

### Official Review · Reviewer_mLKj · 2023-07-20

**Soundness:** 4 excellent
**Presentation:** 3 good
**Contribution:** 3 good
**Rating:** 6
**Confidence:** 4

**Summary:**

The paper considers the random order model originally proposed by Garber et al. '20, and presents a generic framework to take any (approximate) offline algorithm and via usage of coresets turn it into an online algorithm with low average sensitivity, which can be then used to obtain an (approximate) regret bound. Some specific cases are considered where this framework is applied, including online clustering, online low-rank matrix approximation, and online regression.


**Strengths:**

* The high level idea is elegant, the paper is coherent and well written.
* The proposed method offers novel algorithmic approach for batch-to-online reductions.


**Weaknesses:**

* I was missing some context that would allow better understanding the approach presented in the paper; specifically comparison to the approaches considered in [Garber et al., 2020, Sherman et al., 2021]. The role of uniform convergence in coresets construction vs running the offline algorithm directly on the cumulative loss, and average sensitivity vs algorithmic stability. See some of my questions below.
* Perhaps this is not a weakness per se, but (please correct me if I'm wrong - this was not entirely clear to me) the rates obtained do not strictly improve any prior art excluding Section 6 which improves Cohen-Addad et al., '21 but applies less generally. The rest of the results pertain to $\epsilon$-approximate regret while the previous works mentioned obtain standard regret bounds.

**Questions:**

* **Coresets vs direct uniform convergence of the cumulative loss.** At a high level, is it correct to say that the coreset construction is based on uniform concentration of the loss, and that in some specific cases this uniform concentration comes cheap? If so, could we alternatively take the offline algorithm and run it on the cumulative loss as is, then claim a regret bound owed to uniform concentration of the cumulative loss directly? Some more questions related to this point:
	* Does big-$O$ hide a dimension factor in Lemma 5.2?
	* Section 5 presents a coreset construction method, but then in the section that follows (6.2) a different method (that of y Huang and Vishnoi '20) is used. Why?
* **Average sensitivity vs algorithmic stability.** TV is the infimum (over all couplings of the two RVs) of the expected 0-1 distance. This is an upper bound on the Wasserstein distance which is the infimum (over all couplings of the two RVs) of the expected euclidean distance. This Wasserstein distance is basically average algorithmic stability. Is it correct that average sensitivity is a strictly stronger notion?
* **Applicability in the full adversarial setup.** It seems to me the method you propose should also work in a fully adversarial setup. Average sensitivity is akin to uniform stability; the output of your offline algorithm "stabilized" via coresets does not care about stochasticity in the examples. If the output of an approximate ERM does not change much between rounds this should lead to a regret bound, perhaps? Roughly; $\sum_{t} \ell(A(X_t), x_{t+1}) - \ell(A(X_{t+1}), x_{t+1}) \leq \sum_t \beta(t)$ and by "Be-the-Leader" Lemma (Kalai & Vempala '03), $\sum_{t} \ell(A(X_{t+1}), x_{t+1}) \leq \sum_{t} \ell(\theta^\star, x_{t+1})$.
* **$\epsilon$-regret vs competetive-ratio**. How does $\epsilon$-regret compare to competitive ratio from competitive analysis?
* Lines 36-37 "... and when combined with the approximation algorithm ..." - to my understanding, whether the the offline algorithm that operates on the coreset is approximate or exact is irrelevant to the point you are trying to make, here and in the paper more generally, no?



## Minor comments
* "Although the stochastic setting is not often satisfied in real applications, the performance 15 and guarantees of online algorithms in the adversarial case are considerably compromised" - the meaning of this sentence is not clear to me.
* Lemma 4.2: I didn't see the notation "$x = r \pm \beta$" defined anywhere (I gather it signifies $r-\beta \leq x\leq r+\beta$).
* What does it mean to "run $\mathcal A$ with approximation ratio of $(1-\epsilon')$..." (line 4 of Algorithm 3)? Theorem 6.1 does not mention explicitly any requirements from $\mathcal A$. Looking into the detailed version of the algorithm (Algorithm 6), it seems $\mathcal A$ should be a PTAS. Perhaps putting this in the main text could make this a bit clearer.
* ״We remark that the importance of sampling steps in the framework is similar to 193 the ones described in Section 5, which thus allows us to analyze its average sensitivity.״ --> "the importance sampling steps"?

---

> ### Author Rebuttal · Authors · 2023-08-08
>
> We thank the reviewer for the insightful comments! Due to the space limit, we address the major comments here.
> # 1. Comparison to previous works.
> It is possible to convert our results to standard regret bounds. Using the definition of approximate regret, we can rearrange the terms and obtain $\mathbb{E} _{\mathcal{A},\\{x_t\\}}\left[\sum^n _{t=1} \ell(\theta_t, x_t) - \mathrm{OPT}\right] = \epsilon  \mathrm{OPT} + \mathrm{Regret} _\epsilon(n) $. Then by applying the (generalized) AM-GM inequality to the right-hand side and tuning $\epsilon$ appropriately so the inequality holds as equality, we can obtain an exact regret.
> From the approximate regret of our online matrix approximation result, we can translate it to $O(\mathrm{OPT}^{2/3} (d \log n \log(\epsilon^{-1} kn))^{1/3}) $.
> From the approximate regret of our online regression result, we can translate it to $O(\mathrm{OPT}^{2/3} (d \log n \log(\epsilon^{-1} dn))^{1/3})$.  We also want to highlight that our results may be more general in some applications. For online matrix approximation with the random-order setting, the previous result by [Garber et al., 2020] is $O \left(\zeta^{-1} \sqrt{kn}\right)$, where $\zeta$ is the smallest difference between two eigenvalues of $\boldsymbol{A} \boldsymbol{A}^\top$. In contrast, our result does not depend on this $\zeta$. For online regression, [Garber et al., 2020] give $O(\sqrt{n})$-type regret when the matrix $\boldsymbol{A}$ has a small condition number. In comparison, our result maintains no requirement on the condition number.
> # 2. Coresets vs direct uniform convergence of the cumulative loss.
> We agree with the reviewer's comment on a high level. Yet the coreset construction method is still different from the uniform concentration of loss on the technical details because the former guarantees that the multiplicative error is small whereas the latter usually refers to guarantees on additive error. In the case where the offline algorithm is approximate, optimizing the cumulative loss at each step may not lead to small regret. The following shows that at least the be-the-leader lemma does not work. Suppose that at each time step, we run an offline algorithm to obtain an $(1+\epsilon)$-approximate solution on the cumulative loss. Let $\theta^\ast = \min_\theta \sum^t_{i=1} \ell(\theta, x_i)$, we obtain $\theta_t$ such that $\sum^t_{i=1} \ell(\theta_t, x_i) \leq (1 + \epsilon)\sum^t_{i=1} \ell(\theta^\ast, x_i)$. Note that we assume that one has access to $x_t$ before making the decision at time $t$, in our setting one does not even have this.
> Then the regret is
> $$\left(\ell(\theta_1, x_1) + \ldots + \ell(\theta_T, x_T)\right) - (1+\epsilon)\left(\ell(\theta^\ast, x_1) + \ldots + \ell(\theta^\ast, x_T)\right) \leq \left(\ell(\theta_1, x_1) + \ldots + \ell(\theta_T, x_T)\right) - \left(\ell(\theta_T, x_1) + \ldots + \ell(\theta_T, x_T)\right) $$
> Because $\{\theta_t\}^T_{t=1}$ are approximate solutions, we cannot proceed the be the leader lemma as we cannot say
> $\left(\ell(\theta_T, x_1) + \ldots + \ell(\theta_T, x_{T-1})\right) \geq \left(\ell(\theta_{T-1}, x_1) + \ldots + \ell(\theta_{T-1}, x_{T-1})\right) $.
> In the case where the offline algorithm is exact, optimizing with respect to the cumulative loss at each step (follow the leader) works in the stochastic setting. Yet it remains unclear to us whether this result still holds in the random order setting.
>
> Related questions: No, Lemma 5.2 is independent of dimension $d$. The coreset construction method required to derive Theorem 5.1 is obtained by applying Algorithm 2 and the coreset construction proposed by Huang and Vishnoi '20 together to obtain dimension-independent results. The overall algorithm is given in the appendix (Algorithm 7). We will revise this in the final version to make this more explicit.
>
> # 3. Average sensitivity vs algorithmic stability.
> The average sensitivity cannot be directly compared to the algorithmic stability. The algorithm stability describes the change in loss functions by removing or replacing a data point while the average sensitivity describes the change in the algorithm's output by removing a data point. For more discussion on this, we refer to Varma, N., \& Yoshida, Y. (2021). Average sensitivity of graph algorithms. SODA.
>
> # 4. Applicability in the full adversarial setup
> We do not think the argument applies to the adversarial setting, as $\sum_{t} \ell(A(X_t), x_{t+1}) - \ell(A(X_{t+1}), x_{t+1}) \leq \sum_t \beta(t)$ does not hold in the adversarial setting. This is because we can only guarantee that on average, removing a point does not change $X_{t+1}$. Yet for every $t$, removing $x_{t+1}$ might be the worst change to $X_{t+1}$.
>
> # 5. $\epsilon$-regret vs competetive-ratio
> The $\epsilon$-regret quantifies the cumulative discrepancy between the losses incurred by an online algorithm and the $\epsilon$-approximation of the losses obtained by the optimal hindsight solution. The competitive ratio determines the maximum factor by which the online algorithm's cost could exceed the cost of an optimal offline solution in the worst case. The optimal solution is also defined differently, in regret analysis, we consider $\min_\theta \sum_t \ell(\theta,x_t)$ as the optimal value, but in competitive analysis, we consider $\sum_t \min_\theta \ell(\theta,x_t)$ as the optimal value. In general, $\epsilon$-regret and the competitive ratio are not interchangeable.
>
> # 6. Line 36-37
> The approximation ratio of the coreset is irrelevant to the point we are trying to make, but we do need to operate the offline algorithm on the obtained coreset (which enjoys low average sensitivity) to have the overall algorithm enjoy low average sensitivity, and hence low regret.
>
> # Misc
> "the importance sampling steps": We are referring to the importance sampling (based on sensitivity scores) mentioned in Section 5 in the overall coreset construction algorithm for clustering (which is in the appendix, Algorithm 7).

---

> > ### Comment · Reviewer_mLKj · 2023-08-15
> >
> > I would like to thank the authors for their rebuttal.
> >
> > ### 2. Coresets vs UC
> > Could you please clarify the relation between Algorithm 2 and Algorithm 7?
> >
> > ### 3. average stability vs algorithmic stability
> > Whether removing or replacing a datapoint is not important in the definition of algorithmic stability, and a stronger notion (and the one mostly used in practice) of average stability measures the change in output, not loss function.
> > The difference pointed out in Varma, N., & Yoshida, Y. (2021) is with regards to *uniform* stability. I don't see how it applies to average stability (or uniform stability where the datapoints are averaged over; in average stability, both the input sets and datapoints are averaged over).

---

> > > ### Author Response · Authors · 2023-08-15
> > >
> > > We thank the reviewer again for the comments. We address the raised question in the following.
> > >
> > > # 1. Algorithm 2 and Algorithm 7
> > > Algorithm 2 provides us a template for designing coreset construction algorithm with small average sensitivity by perturbing the weights (Line 5 - 10). Algorithm 7 applies this idea to Huang and Vishnoi [2020]'s algorithm. Specifically, Lines 6 and 11 describe the perturbations to the weights.
> > >
> > > # 2. Average sensitivity vs algorithmic stability
> > > We first want to remark that average sensitivity does not take the average over the input set, but average stability does (we follow definition 4 from Lei, Y., & Ying, Y. (2020) here). If the average sensitivity is bounded by beta, then the average stability is bounded by beta times the maximum distance between $\theta$'s, which is the algorithm parameter (or the loss difference incurred by them, if the average stability is defined wrt to the loss). As average sensitivity consider the worst case (in terms of input set), we do not believe that the average stability is sufficient to derive our results. In the stochastic setting, it is not clear whether average stability would be sufficient. In our current analysis, we crucially used the fact that TV is bounded (so we can say that the outputs are the same with high probability). For average stability to be sufficient, the loss has to be Lipschitz, which may not be always true (for example when the matrix has a high condition number in linear regression). We are also a bit confused about why TV upper bounds the Wasserstein distance (we believe that this is true if data are assumed to be in a unit ball, but we do not think this is true in the general case).
> > >
> > > Lei, Y., & Ying, Y. (2020). Fine-grained analysis of stability and generalization for stochastic gradient descent. In International Conference on Machine Learning (pp. 5809-5819). PMLR.

---

> > > ### Author Response · Authors · 2023-08-20
> > >
> > > We thank the reviewer again for the detailed comments. We hope our response answered your questions, and we will be more than happy to answer any additional questions.

---

> > > > ### Comment · Reviewer_mLKj · 2023-08-20
> > > >
> > > > Sorry for the late response - I would like to thank the authors for their detailed replies. I do not have further questions.
> > > > I was and still am in favor of acceptance, of course.

---

> > > > > ### Author Response · Authors · 2023-08-20
> > > > >
> > > > > Thank you so much for the positive review and review again!

---

### Author Rebuttal · Authors · 2023-08-08

We thank the reviewers for their insightful comments and constructive advice!

We would first like to highlight that our work is primarily theoretical in nature, as our focus lies in exploring and analyzing the theoretical foundations of the general framework proposed. Moreover, the related works are mostly theoretical as well, such as [Cohen-Addad et al., 2021] (for online clustering), [Garber et al., 2020] (for online regression, and online matrix approximation in PCA). Due to the scarcity of available code for baseline algorithms, it is hard to conduct a fair empirical evaluation of our approach in the limited time of the author's response period.

We here provide a preliminary empirical evaluation of our framework in the context of online $k$-means clustering, online linear regression (see the additional file), and various approximation ratios and experimental setup ($\epsilon = 0.1, 0.01, 0.001$, with $k=3$ or $k=5$ clusters). We compare the performance of the proposed algorithm to the hindsight optimal solution. For $k$-means clustering, we obtain the hindsight optimal solution by applying $k$-means++ to all the data. In the context of regression, we utilize the least square formula to compute the hindsight optimal solution. Our experimental results demonstrate that the proposed algorithm is highly effective, and its performance aligns with our theoretical findings.

---

### Decision · Program_Chairs · 2023-09-21

**Decision:**

Accept (poster)

**Comment:**

All five reviewers are positive about this paper and think it should be accepted and I support their decision. Please take into account to reviewers comments when preparing the final version.